# Parahippocampal neurons encode task-relevant information for goal-directed navigation

Alexander Gonzalez*, Lisa M Giocomo*

Department of Neurobiology, Stanford University School of Medicine, Stanford, United States

**Abstract** A behavioral strategy crucial to survival is directed navigation to a goal, such as a food or home location. One potential neural substrate for supporting goal-directed navigation is the parahippocampus, which contains neurons that represent an animal's position, orientation, and movement through the world, and that change their firing activity to encode behaviorally relevant variables such as reward. However, little prior work on the parahippocampus has considered how neurons encode variables during goal-directed navigation in environments that dynamically change. Here, we recorded single units from rat parahippocampal cortex while subjects performed a goal-directed task. The maze dynamically changed goal-locations via a visual cue on a trial-to-trial basis, requiring subjects to use cue-location associations to receive reward. We observed a mismatch-like signal, with elevated neural activity on incorrect trials, leading to rate-remapping. The strength of this remapping correlated with task performance. Recordings during open-field foraging allowed us to functionally define navigational coding for a subset of the neurons recorded in the maze. This approach revealed that head-direction coding units remapped more than other functional-defined units. Taken together, this work thus raises the possibility that during goal-directed navigation, parahippocampal neurons encode error information reflective of an animal's behavioral performance.

*For correspondence:
alexg8@stanford.edu (AG);
giocomo@stanford.edu (LMG)

## eLife assessment

In this study, neurons were recorded and combined across the parahippocampal area while rats performed a memory-guided spatial navigation task. Sophisticated analytical tools were used to provide **convincing** evidence that neuronal populations in these areas show behavior-related changes that might indicate the encoding of errors by the system. The **valuable** results suggest that rate remapping is a likely mechanism to support changes in representations that support memory-guided behavior in these regions, most interestingly in neurons that code head direction.

## Introduction

Navigation to a goal location, such as a food source or home, is a behavior central to the survival of many species. This behavior begins with a serial transformation of sensory inputs and ends in a sequence of motor actions reflective of a movement decision. Between sensation and motor action, several parahippocampal regions represent variables theorized to support the computations needed for goal-directed navigation. These regions include the medial entorhinal cortex (MEC) and pre- and para-subiculum (PrS, PaS), which contain functionally defined neurons that encode an animal's spatial position, orientation and movement through the external environment (*Hafting et al., 2005*; *Kropff et al., 2015*; *Solstad et al., 2008*; *Sargolini et al., 2006*; *Høydal et al., 2019*; *Boccara et al., 2010*). Additionally, the PrS and PaS project to the MEC, which is reciprocally connected to the hippocampus

(*Cappaert and Witter, 2015*), a region strongly implicated in supporting memory and navigation (*Ekstrom and Ranganath, 2018*; *Buzsáki and Moser, 2013*; *Lisman et al., 2017*). However, much of the prior work characterizing functionally defined neurons in MEC, PrS, and PaS has considered neural activity recorded as rodents forage for random food rewards in open fields environments. One limitation inherent to such open field foraging behavioral data sets is the lack of pre-defined decision points, which creates challenges to understanding the relationship between neural activity and goal-directed navigation. While several studies have attempted to address these challenges by utilizing linear environments, these works typically employed alternation tasks that lack experimental control over decision making junctures (*Frank et al., 2000*; *Lipton et al., 2007*). Thus, how parahippocampal neurons encode navigational variables during goal-directed navigation in complex environments remains incompletely characterized.

Goal directed navigation encompasses both spatial and non-spatial coding elements, as an animal must estimate its current body position, head orientation, and running speed and integrate knowledge of behaviorally relevant goals or sensory cues. Both the MEC and PaS contain large populations of neurons that encode navigationally relevant variables for computing an animals spatial position, including: body position, head direction and running speed. Individual MEC neurons often exhibit a mixture in their selectivity for encoding these variables and, while ~50% of MEC neurons can be classified based on metrics that quantify the firing rate profile along a behavioral dimension (tuning curves), a larger proportion of neurons can be classified using more unbiased statistical approaches (*Hardcastle et al., 2017b*; *Diehl et al., 2017*; *Spalla et al., 2022*). In the PaS, ~50% of neurons code for heading direction (*Boccara et al., 2010*; *Tang et al., 2016*), with coding for position or running speed present in about ~40% neurons (*Spalla et al., 2022*; *Tang et al., 2016*). MEC has also been shown to contain neurons that encode variables related to non-spatial features of a task or behavior. For example, subsets of MEC neurons encode time elapsed during immobility (*Heys and Dombeck, 2018*), change their firing rate or the spatial location at which they are maximally active in response to reward locations (*Butler et al., 2019*; *Boccara et al., 2019*), and encode an animal's position in task-relevant space during a non-spatial auditory frequency task (*Aronov et al., 2017*).

In addition to encoding navigationally relevant variables, parahippocampal regions have been proposed to indirectly support memory through the reciprocal projections to the hippocampal formation (*Buzsáki and Moser, 2013*). Depending on the task, goal-directed navigational behaviors might require knowledge of remembered cue-reward associations or memory of previous rewarded or behaviorally significant spatial locations (*Robinson et al., 2020*; *Gauthier and Tank, 2018*; *Hollup et al., 2001*). At the level of neural activity, these types of memory associations may be facilitated by changes in neural activity that occur in response to changes in environmental sensory cues, such as new visual landmarks, or the presence of a reward (*Colgin et al., 2008*; *Whittington et al., 2020*). For example, in the hippocampus, place cells that are active in one or a few spatial locations will move the spatial position at which they are maximally active across different open-field environments, a phenomenon referred to as 'global remapping', or more specifically 'location remapping'. Place cells can also change their firing rate, a phenomenon referred to as 'rate remapping', which can occur with global remapping or independent of global remapping when changes to the environment are small (*Colgin et al., 2008*; *O'Keefe, 1976*; *Leutgeb et al., 2005*; *Plitt and Giocomo, 2021*). In the hippocampus, both global and rate remapping have been observed in response to changes in environmental sensory cues, such as the color or shape of the spatial environment, and the presence of reward (*Muller and Kubie, 1987*; *Bostock et al., 1991*; *Knierim et al., 1998*). Remapping events observed in the hippocampus have been shown to occur in concert with changes MEC spatial firing patterns, at both the correlational level (place cell remapping correlates with re-orientation of MEC grid-cells; *Fyhn et al., 2007*), and at the causal level (inactivation of MEC coincides with hippocampal global remapping; *Miao et al., 2015*). Both global and rate remapping measurements provide, at the population level, a quantification of stability in the neural representation of an environment through spatial correlations (*Colgin et al., 2008*). However, remapping might be indicative of flexibility in the representation of space, other behaviorally relevant variables, or experience (*Plitt and Giocomo, 2021*). Thus, quantification of remapping can be experimentally leveraged to examine whether spatial representations flexibly adapt to behaviorally relevant factors such as distinct cue-reward associations that guide decisions made during goal-directed navigation.

Neurons in MEC also exhibit global and rate remapping between different spatial environments. Moreover, MEC neurons also remap in response to variables that flexibly change during goal-directed navigation, including: reward locations, running speed and an animal's trajectory. The presence or change in the position of an unmarked remembered reward location can evoke both rate and global remapping in position and orientation encoding cells in MEC (*Butler et al., 2019*; *Boccara et al., 2019*). Changes in an animal's running speed have been observed to correlate with both rate remapping (*Hardcastle et al., 2017a*; *Bant et al., 2020*) and global remapping in MEC (*Low et al., 2021*). In structured linear track mazes, in which animals perform spatial alternation tasks, MEC exhibits trajectory-dependent remapping for the same spatial location (*Frank et al., 2000*; *Lipton et al., 2007*; *O'Neill et al., 2017*). However, while previously observed trajectory dependent remapping is indicative of prospective coding during stereotyped behavior (e.g. alternation), it remains unknown how MEC, PaS, and PrS spatial patterns change in unstructured tasks (e.g. random trial-to-trial structure), or with experimenter-controlled cues and goals in the same environment. Moreover, while work in the hippocampus has shown that the degree of trajectory-dependent remapping correlates with behavioral performance (*Kinsky et al., 2020*), and causal evidence revealed that shifted spatial representations correspondingly shifts the expected location of reward (*Robinson et al., 2020*), such neural-behavior relationships have not been examined in the upstream MEC, PaS, or PrS.

Here, we examined single unit neural activity in MEC, PrS, PaS as rats performed a visually cued navigational task that incorporated both goal-directed navigation and random foraging. We hypothesized that neurons in the parahippocampal formation would code for behaviorally relevant variables, like cue identity and reward. Further, we hypothesized that this neural coding could relate to the subjects performance in the task as the parahippocampal region is upstream of the hippocampus which has been shown to exhibit those relationships. These hypotheses were confirmed when we found that a large percentage of neurons changed their firing activity as a function of cue identity and reward, and that the extent of separability (as measured through remapping) in firing activity correlated with behavioral performance. To further understand how neurons changed their spatial activity patterns, we employed encoding models of rate and global remapping, where we found that changes in neural activity in the task were sufficiently explained by rate remapping. Interestingly, neural representations were also separable contingent on the presence or absence of reward at the goal well. Finally, single-unit recordings during open field foraging revealed that neurons that encoded head-direction remapped more strongly than other functionally defined units. Together, these results point to the MEC and upstream PrS/PaS as nodes in the transmission of behaviorally relevant variables during goal-directed navigation.

## Results

### Tree-Maze behavior and electrophysiological recordings

To examine whether MEC, PrS and PaS neurons encode variables related to goal directed navigation, we designed a linear maze with two decision junctions, referred to here as the Tree-Maze (*Figure 1*, *Figure 1—figure supplement 1*, see *Ainge et al., 2007* and *Joo et al., 2021* for similarly structured mazes). Rats (n=5) learned to navigate a sequence of wells to receive liquid milk rewards (*Figure 1a*). There were four possible Goal-wells (G1-G4), and subjects received double rewards for triggering the correct Goal-well. Trials started at the Home-well (H). After animals triggered the Home-well (H), a visual cue turned on that indicated the correct branch to navigate after the Decision-well (D) (RC = Right Cue, purple, Goal-wells G1-G2; LC = Left Cue, green, Goal-wells G3-G4) (*Figure 1b*). For the correct branch, only one Goal-well contained reward, which was randomly chosen on each trial. Thus, the task combined goal directed navigation (selecting the correct branch) with random foraging (selecting the first Goal-well to visit). After navigating correctly, or incorrectly (no reward and a 10 s timeout), the subject had to navigate back to the Home-well to initiate a new trial. The presence of two possible goal locations after the decisions kept subjects engaged, and dissociated the decision from the spatial location of the reward. For analyses, we segmented the maze into zones (*Figure 1a*), such that the location of the subject at any given time-point could be localized to a single zone (*Figure 1c*).

We trained 5 rats on the Tree-Maze task (see Methods-Behavioral Training). Subjects took an average of 25 sessions to reach the performance criterion for surgery (# sessions to criterion, range

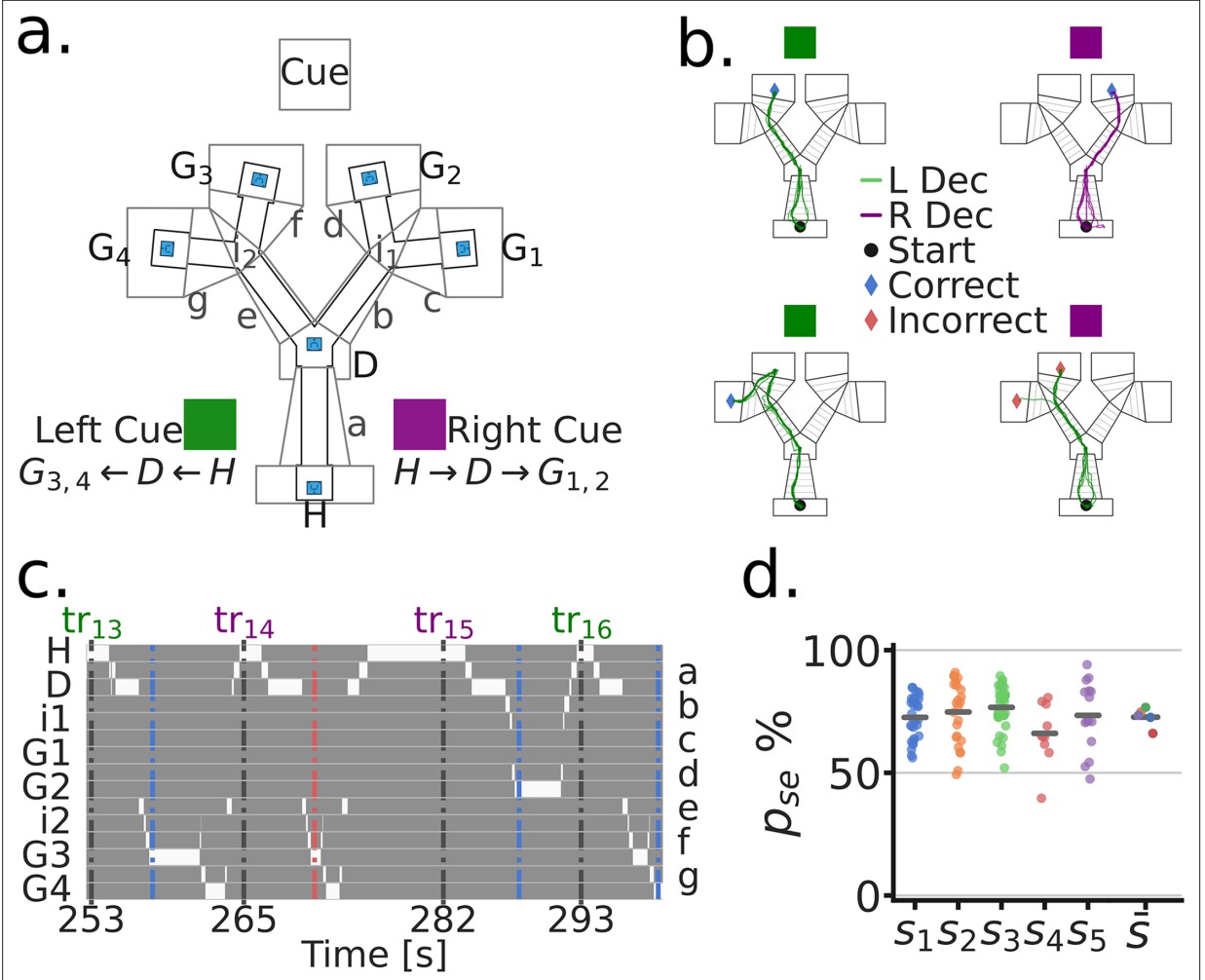

**Figure 1.** Goal directed navigation on the Tree-Maze task. (**a**) Top view of the Tree-Maze layout and segmentation (indicated by upper and lower case letters) used for analyses (height = 1.4 m, width = 1.2 m). The LED cue panel was located at the end of the maze (flattened for illustration). Colored cue cards on the side of the maze indicate the two cue types by trial (Purple = Right Cue [RC], Green = Left Cue [LC]). Possible reward locations denoted by capital letters (Home = H, Decision = D, Goal = G1 G4). Lower case letters correspond to the segments shown on the right side of panel c. Reward wells highlighted in blue. (**b**) Colored lines indicate example trajectories of a rat, five trials per panel. L Dec indicates a trajectory towards the left branch and R Dec indicates a trajectory towards the right branch. Left column (Left/Green cue), trajectories to G3 (top) and G4 (bottom) for reward. Both of these were correct navigational decisions and were rewarded. Right column (Right/Purple cue), trajectories to G2 (top) and G3 (bottom), only the top trajectories resulted in reward at a goal. (**c**) Binary trajectory segmentation time window. At any given time-point, the subject can only be at one location in the maze (indicated by the white bins). Bottom axis indicates trial start times (seconds). Top axis indicates the trial number (tr), colored coded by cue. Left/Right axis indicates the identity of the segment, lower case letters on the right correspond to those in panel a. Blue and red dashed vertical lines indicate the end of a trial (blue = correct; red = incorrect). (**d**) Task performance by subject. Each dot corresponds to a session by subject. For $\bar{s}$, dots indicates the subject mean. $p_{se}$ = performance in a session.

The online version of this article includes the following figure supplement(s) for figure 1:

**Figure supplement 1.** Data collection and behavior apparatus.

**Figure supplement 2.** Subject learning and post-surgery behavior.

**Figure supplement 3.** Sagittal histology sections illustrating the location of recording electrodes.

**Figure supplement 4.** Number of units by session and subject.

= 11–50; criterion: accuracy ≥ 70% and # trials ≥ 80; *Figure 1—figure supplement 2*). Post-electrode implantation (Histology, *Figure 1—figure supplement 3*, # units by subject *Figure 1—figure supplement 4*), criteria for inclusion in the analyses were: # units ≥ 1, # trials ≥ 50 (across subject median accuracy = 72.6%, median number of trials per session = 110; *Figure 1d*). The cue on each trial was random, which allowed us to keep track of performance on Switch trials (in which the cue on trial N+1

was different than the cue on trial N, *Figure 1—figure supplement 2*). Indicating that subjects were adept at the task and not simply following an alternation or exploitative strategy, the mean Switch trial performance was similar to non-Switch trials (Switch trial median accuracy = 76.9%; median number of Switch trials per session = 54). Together, the behavioral requirements of the task and structure of the Tree-Maze allowed us to then examine whether neural activity corresponded to different task dimensions (e.g. cue, decision, reward, spatial position).

## Visual cues evoked spatial remapping

To investigate the effect of the cue on the spatial firing profiles of single units (i.e. neurons), we generated spatial maps by cue condition for outbound trajectories (H-well to G-well, *Figure 2a*). For each unit, the spatial locations of spikes along trajectories were used to obtain trial summary metrics for each maze segment as a function of cue (*Figure 2b*; *Figure 2—figure supplements 1–3*). Across trials, the difference on cue coding by maze segment was quantified by the Mann-Whitney Z transformed U statistic ($U_Z$), (*Figure 2b and c*; *Figure 2—figure supplement 4* and *Figure 2—figure supplement 5*). In the scenario in which the cue had no effect on spatial coding, the distributions of $U_Z$ scores will be close to zero for all maze segments (*Figure 2c*). On the other hand, the statistic will be negative if firing rates for the Left-Cue (LC) trials were greater than for Right-Cue (RC) trials, and positive if the firing rates for the RC trials were greater than for the LC trials. Using a linear mixed effects model we found a main effect of Segment with follow-up pairwise comparisons showing a left >stem > right pattern (LMEM; accounted for within subject variance and repeated measures for each session, Segment Likelihood Ratio Test $LRT = \chi^2_3 = 85.25$, $p = 3.07e^{-19}$; joint hypothesis test of left >stem > right Wald Test $\chi^2_2 = 90.45$, $p = 2.27e^{-20}$). Hence, across the population there was a significant bias for RC trials to be associated with greater firing rates on the left segment of the maze (relative to LC trials), and for LC trials to be associated with greater firing rates on the right segment of the maze (relative to RC trials). In both of these instances, the increased firing rate thus follows an incorrect decision, that is, trajectories that did not lead to a reward were the trajectories associated with an increased firing rate.

Next, to quantify the degree of spatial firing rate changes, we generated spatial zone rate maps as a function of cue. Importantly, these maps were generated from balanced re-sampling of correct and incorrect trials (see Methods - Behavioral Training). Concretely, the LC maps were generated by sampling an equal number of correct and incorrect trials from all the trials that had a LC (*Figure 2b*). A distribution of scores was computed by taking the Kendall correlation, $\tau$, between LC and RC spatial maps for each trial re-sampling instance (*Figure 2b*). An appropriate null distribution was generated by sampling equal numbers of LC and RC trials from even and odd trial sets, computing the spatial correlation, and then comparing the resulting null correlation score to the cue correlation scores to generate a remapping score $\bar{z}_{\Delta\tau}$. Across units we observed a significant negative shift in $\bar{z}_{\Delta\tau}$ (LMEM: mean = $-0.67$, $p = 0.003$, $CI_{95\%}=(-1.105,-0.225)$) (*Figure 2d*). This negative shift indicated that the cue induced a significant level of remapping in firing patterns when compared to a null distribution.

## Remapping of spatial representations relates to greater task performance

Given that trajectories after incorrect decisions tended to elicit an elevated FR (*Figure 2c*), we hypothesized that the magnitude of the cue remapping score would related to a subject's performance in the task. To examine this, we related the cue remapping score of each unit $\bar{z}_{\Delta\tau}$ to the corresponding task performance for the recorded session $p_{se}$ (*Figure 2e*). There was a significant negative relationship between performance accuracy on a given session $p_{se}$ and the degree of cue remapping calculated for single units $\bar{z}_{\Delta\tau}$ (Kendall's $\tau = -0.27$; LMEM: $\bar{z}_{\Delta\tau}$ slope = $-0.12$, $CI_{95\%} = (-0.15,-0.09)$, $LRT = \chi^2_1 = 51.09$, $p = 8.84e - 13$). Given that this effect was observed for the majority of units, we hypothesized that the effect should be reflected in the mean of the cue remapping scores for co-recorded units in a session $\bar{z}_{\Delta\tau\,\bar{se}}$ (*Figure 2f*). Indeed, when considering co-recorded units in a session, the correlation between cue remapping and behavioral performance was even more pronounced (Kendall's $\tau = -0.41$; LMEM: $\bar{z}_{\Delta\tau\,\bar{se}}$ slope = $-0.22$, $LRT = \chi^2_1 = 14.34$, $p = 1.4e - 4$, $CI_{95\%}=(-0.33,-0.11)$). Population level correlations showed the same effect, (*Figure 2g*). At the individual subject level, this correlation between performance accuracy and cue remapping was observed in all but one subject, which had a small number of units (*Figure 2h*). These same effects were observed on multiple

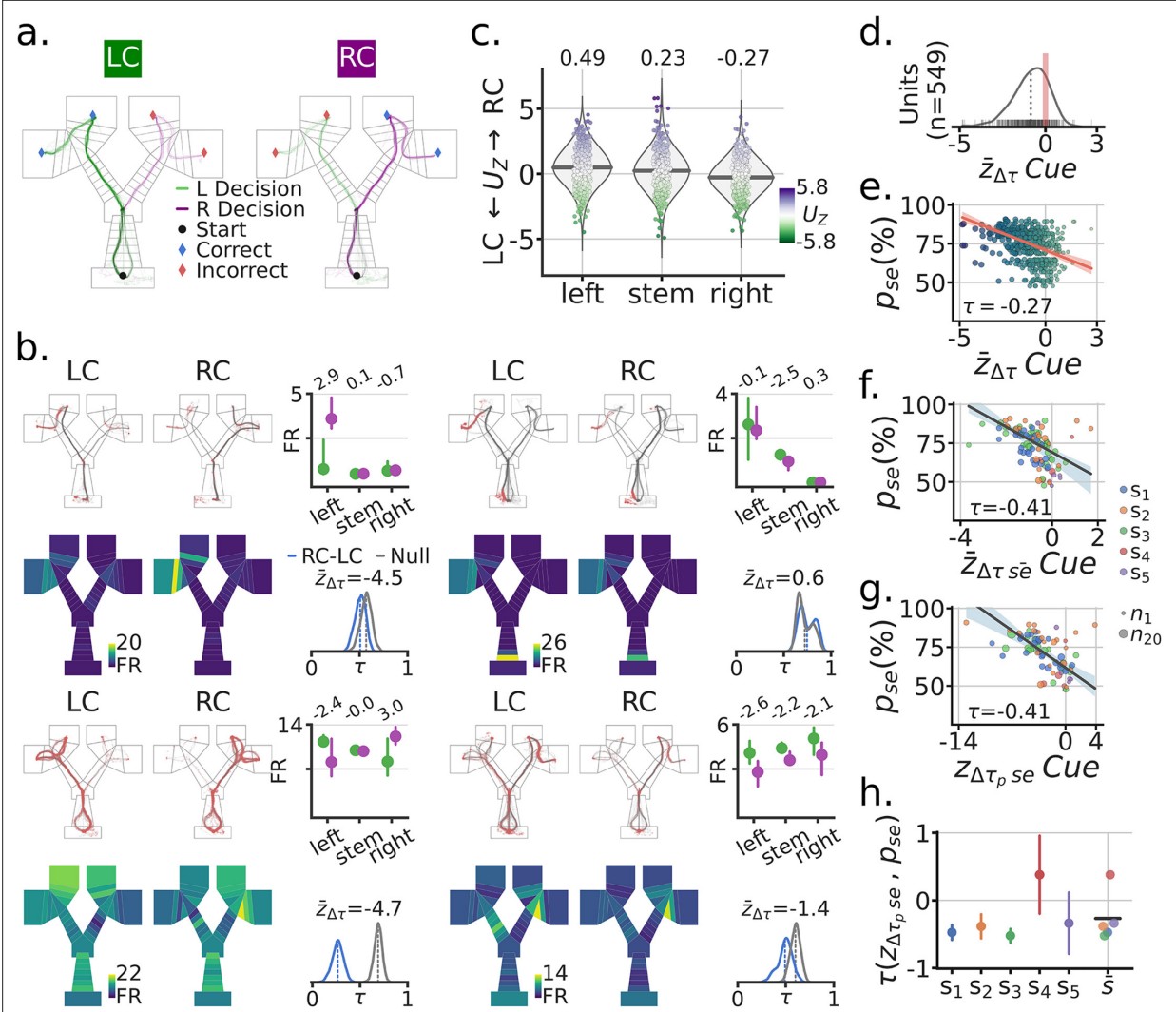

**Figure 2.** Spatial remapping was associated with the visual cue and correlated with task performance. (**a**) Example session trajectories for all LC (Left-Cue) and RC (Right-Cue) trials. (**b**) Four example single-units. For each example unit: Top row, left and middle, outbound trajectories separated by cue (red dots = spikes). Top-right, trial median activity by cue and maze segment (left, stem, right), numbers at the top indicate the Mann-Whitney Z transformed U statistic for the difference between the RC and LC trial distributions $U_z$. Bottom row, mean spatial rate maps by zone and cue, color coded for minimum (firing rate [FR]=0, blue) and maximum (yellow) values. Bottom right, re-sampled distribution of correlations between RC and LC maps in blue for that unit, and in grey the corresponding null distribution. Remapping score $\bar{z}_{\Delta\tau}$ is the mean remapping score for each unit. (**c**) Distributions of $U_Z$ scores for all recorded units by maze segment. Purple means higher FR for RC than LC, Green means higher FR for LC than RC. Note the higher FR for RC on the left segment (far left) and higher FR for the LC on the right segment (far right). (**d**) Distribution of mean remapping scores $\bar{z}_{\Delta\tau}$ by unit, note the negative shift in the distribution of scores. (**e**) Scatter-plot between the task performance on a given session $p_{se}$ and the cue remapping scores $\bar{z}_{\Delta\tau}$ for recorded units in that session. Size and color of dots scale with the x axis for illustration. Regression line in red with a $CI_{95\%}$ band. Kendall correlation score between behavior and remapping score shown. (**f**) Scatter-plot between $p_{se}$ and the mean remapping score across units recorded in a given session $\bar{z}_{\Delta\tau\,\bar{se}}$. Size of dots indicate number of co-recorded units, color codes correspond to different subjects. Regression line and corresponding $CI_{95\%}$ band shown in grey. (**g**) Like (**f**) but with neural population correlation, composed of the spatial rate maps for all recorded units in a session. (h) Correlation between $p_{se}$ and $\bar{z}_{\Delta\tau\,\bar{se}}$ by subject, with bootstrapped standard deviation (B=500). $\bar{s}$ is the across subject mean.

The online version of this article includes the following figure supplement(s) for figure 2:

**Figure supplement 1.** Example rate maps of segment selective units.

**Figure supplement 2.** Spatial rate maps of cue selective units.

**Figure supplement 3.** Unit examples of stronger cue coding while in the incorrect branch.

**Figure supplement 4.** Incorrect coding in mean firing rates.

**Figure supplement 5.** Incorrect coding in Z-scored firing rates.

*Figure 2 continued on next page*

**Figure supplement 6.** Cue remapping vs behavior control analyses for isolated units.

**Figure supplement 7.** Cue remapping vs behavior control analyses for isolated units and MUA.

versions of these analyses (*Figure 2—figure supplement 6, 7*). Together, these results demonstrate cue-based remapping of parahippocampal spatial firing patterns in a goal directed navigation task, with the extent of remapping significantly correlating with behavioral accuracy in the task.

Next, we sought to understand the extent and the manner in which the visual cue was encoded in the FR of individual units during the task. We compared three different linear encoding models: (1) a zone/position model $Z$, (2) a cue rate-remapping model $Z + C$, and (3) a cue global-remapping model $Z \times C$. The zone only model $Z$, simply predicted a neuron's FR based on the zone/position the subject was occupying at a given time. The cue rate-remapping model $Z + C$, also received a cue identity input, acting as gain on the model to uniformly modify the predicted FR across all zones as a function of cue (*Figure 3a and b*). The cue global-remapping model $Z \times C$ doubled the number of input zones, such that only one set of zones could be active on a given cue trial. This approach thus modeled the possibility of a complete re-shuffle of the spatial firing patterns as a function of cue (*Figure 3—figure supplement 1*). For each unit, the cross-validated coefficient of determination $R^2$ was used to asses model performance (individual scores are shown in *Figure 3c*). Note that a negative $R^2$ indicates that

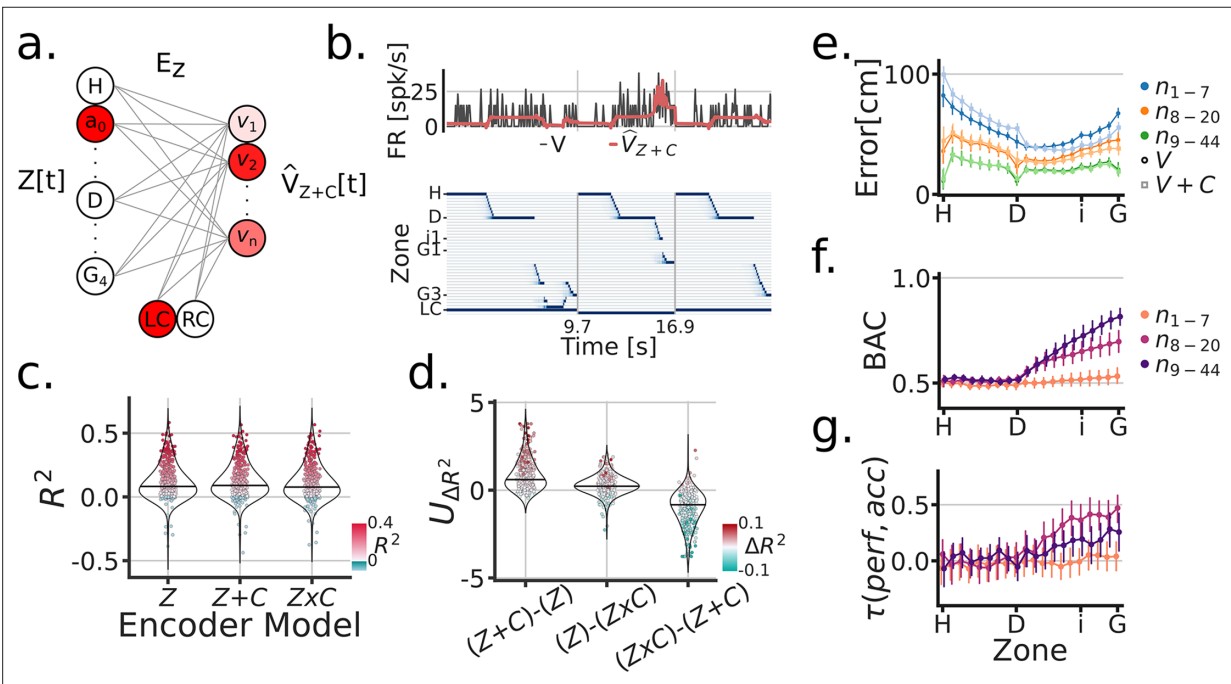

**Figure 3.** Cue modeling revealed rate remapping and trial-wise correlations to behavior. (**a**) Linear zone encoding model with cue $Z + C$, at a given sample time $t$, the current position of the animal and the cue identity is multiplied by learned weights to predict FR for each recorded unit $\hat{V}_{Z+C}$. (**b**) Example time window of $Z[t]$, the true FR in black $V[t]$ and the predicted FR in red $\hat{V}_{Z+C}[t]$. (**c**) Model comparison between three types of zone encoding: $Z \rightarrow$ only zones, $Z + C \rightarrow$ zones + cue, $Z \times C \rightarrow$ a set of zones for each cue. Each dot is a unit, blue dots were negative $R^2$, red scales with $R^2$ and y-axis. (**d**) Model comparison scores. Y-axis is the Mann-Whitney Z transformed statistic for comparing the $R^2$ on test folds. Colorbar indicates the median difference in $R^2$ across test folds. (**e**) Performance of linear zone decoder models $V$ (circles) and $V + C$ (squares). Y-axis is the error distance in cm between the predicted and true zone, X-axis is the linearized Tree-Maze zones displayed as H (Home-well) to D (Decision-well) to i (second intersection/branching) to G (Goal-well). Linearization achieved through averaging the equivalent trajectories towards the goal. The hue shade provides groupings of sessions according to number of co-recorded units (both isolated an MUA included in these analyses). (**f**) Performance of linear decision decoder by zone, with color indicating number of units. Note the sharp decision well split in the performance. BAC = balanced accuracy. (**g**) Correlation between the subject's performance and the model by zone. Color groupings as in (**f**). Model performance is the comparison between the output of the decision decoder and the true identity of the cue, the same computation used to assess a subject's performance.

The online version of this article includes the following figure supplement(s) for figure 3:

**Figure supplement 1.** Encoder and decoder model diagrams.

the mean firing rate of the neuron explained more activity variance than the fitted model (on held-out test data). A subtle but consistent difference in model scores was apparent between the rate- and global remapping model (*Figure 3c and d*), such that the rate remapping model outperformed the global remapping and position only model (LMEM: $U_{\Delta_{R^2}}$ Model $LRT = \chi_3^2 = 962.05$, $p = 1.24e^{-209}$) (*Figure 3d*). The overall single unit scores showed a mean $R^2$ of 12% for the $Z + C$ model.

To further understand how the encoding of the cue across units could modulate the strong position coding present in parahippocampal neurons, we trained a linear decoder to predict the position of the subject using the neural population activity patterns with and without the identity of the cue ($V + C$ and $V$ models, respectively, *Figure 3e*). On held-out test data, we observed significant modulation with the cue as an input, reflected by decreased errors (the distance between the predicted and true position), but only in sessions with high number of units (LMEM: interaction between model and units groupings: $LRT = \chi_2^2 = 17.04$, $p = 1.9e^{-4}$). Additionally, the addition of cue information was most beneficial in reducing errors post-decision (LMEM: triple interaction of model, unit grouping and pre-/post-decision marker: $LRT = \chi_2^2 = 214.31$, $p = 2.9e^{-47}$). Next, we trained a classifier to predict the Left/Right decision of the subject using the population activity (*Figure 3f*). Prospective decoding of the decision prior to the decision point (stem) was at chance, while it steadily increased beyond the decision point across sessions, more so in sessions with more than 8 units (LMEM: interaction of unit groupings and pre-/post decision marker, $LRT = \chi_2^2 = 356.82$, $p = 3.28e^{-78}$). The decoding of decision is reflective of the strong position coding in parahippocampal units, and also indicative of the behaviorally relevant information content of these neurons. An important note is that the number of units in a session heavily influenced the ability of the decoder to predict an upcoming Left/Right decision in the stem. Concretely, 19/42 sessions had above chance decoding with 8 or fewer co-recorded units, while 23/36 sessions with 20 or more co-recorded units had above chance decision decoding. Finally, because the decoder has trial-wise predictions of an animal's decision Left/Right, we tested the extent to which parahippocampal representations matched the subject's behavior (*Figure 3g*). We compared decoder outputs (Left/Right decisions) to the identity of the cue, thus providing a measure comparable to the performance of subjects on the task. With this approach, we relate the performance of the subject to the model's performance with a LMEM, with unit groupings and pre-/post-decision marker as covariates, and found a triple interaction between these predictors ($LRT = \chi_2^2 = 43.48$, $p = 3.61^{-10}$). The interaction can be broken down as follows: post-decision decoder predictions related to the subject's performance and was modulated by the number of co-recorded neurons. These analyses demonstrate that the parahippocampal representations are not only reflective of behavior on the mean sense, as assessed through spatial remapping, but also at the individual trial-level.

## Absence of reward leads to higher activity and spatial remapping

After the subjects navigated to Goal-wells, they received reward contingent on whether they made the correct navigation decision. Inbound trajectories from the Goal-wells to the Home-well could thus be separated by rewarded (correct trials) versus unrewarded (incorrect trials) (*Figure 4a*). Based on previous data (*Butler et al., 2019*; *Boccara et al., 2019*), we posited that reward could also elicit firing rate changes in the spatial representations of parahippocampal neurons. We quantified firing rate changes for each unit as a function of reward by maze segment (*Figure 4b and c*; *Figure 4—figure supplements 1 and 2* as a control). Across units, we found that unrewarded trials tended to have elevated firing rates across all maze segments (proportions: left = 386/565, stem = 340/565, right = 412/565; LMEM: segment $LRT = \chi_3^2 = 47.70, p = 7.91e^{-12}$).

Given the firing rate differences observed between rewarded and unrewarded trials, we expected to see differences in position coding for the spatial maps corresponding to these trials sets. For each unit, we generated a distribution of correlations between rewarded and unrewarded inbound spatial trajectories (bootstrapped trials with equal numbers of LC and RC). The null distribution was then the correlations between even and odd inbound trial trajectories (bootstraps having an equal number of rewarded and unrewarded trials). As with the cue, we quantified the difference between these distributions for each unit to obtain a remapping score $Z_{\Delta_\tau}$, *Figure 4b and d*. Across units, we observed a significant negative shift of the remapping score, which indicates that spatial maps changed based on the receipt or absence of reward $\bar{z}_{\Delta_\tau}$ (LMEM: mean $= -1.03$, $CI_{95\%} = (-1.6, -0.46)$, $p = 4.1e^{-4}$) (*Figure 4d*).

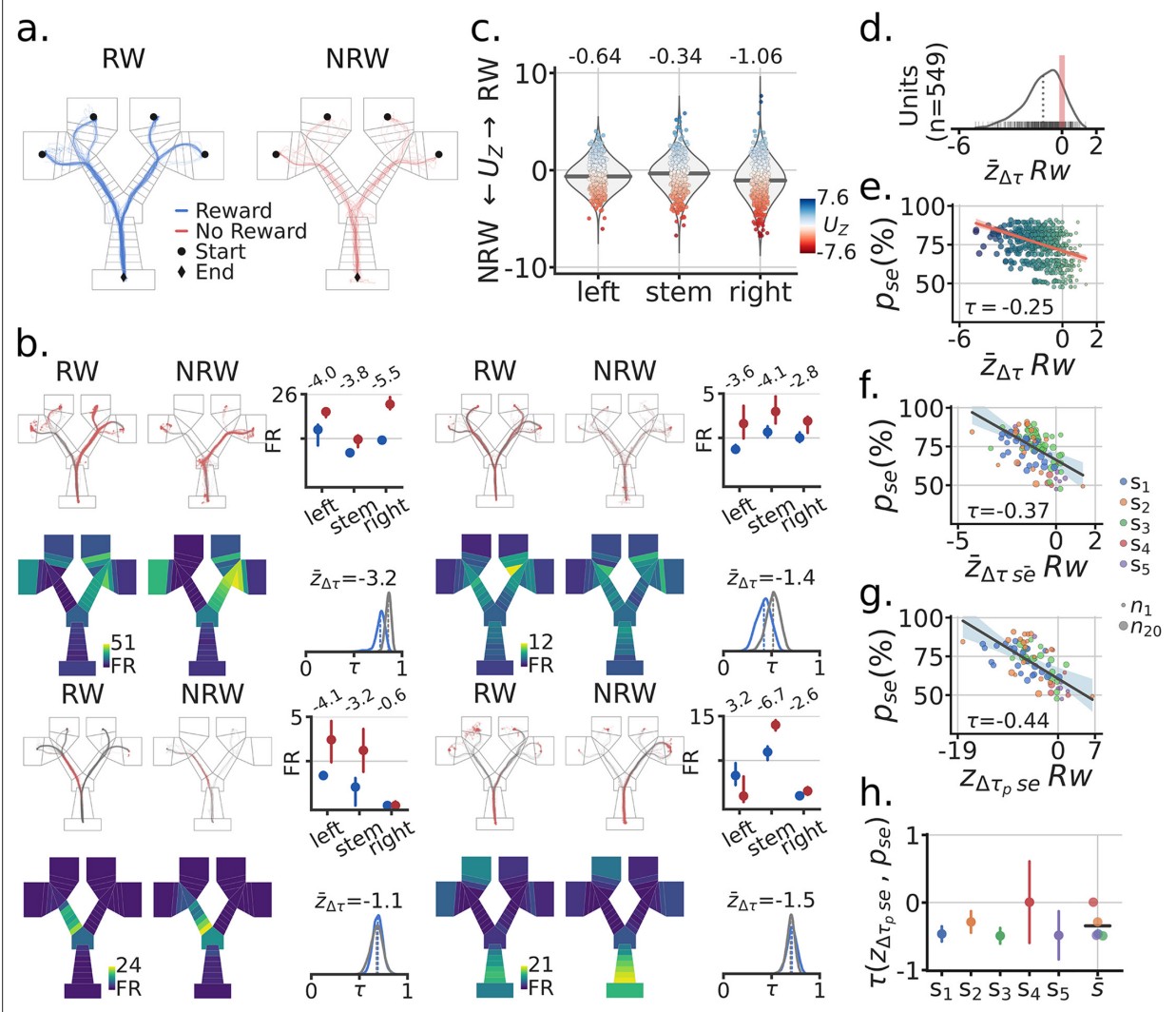

**Figure 4.** Absence of reward leads to higher activity and spatial remapping. (**a**) Example session RW (reward) and NRW (no-reward) trials. (**b**) Four example units (**i–iv**). Top rows, inbound trials trajectories by RW/NRW (red dots = spikes). Top-right, trial median activity by RW and segment in the maze. Firing rate difference score $U_z$ as described in the main text. Bottom rows, mean spatial rate maps by zone and RW, minimum FR = 0. Bottom right, re-sampled distribution of correlations between RW and NRW maps in blue for that unit, and in grey the corresponding null distribution. Remapping score $\bar{z}_{\Delta\tau}$ is the mean remapping score for each unit. b.i Session is the same as in panel a., other units from different sessions. (**c**) Distributions of $U_Z$ scores for all recorded units by maze segment. Blues means higher FR for RW than NRW trials, Red higher NRN than RW. (**d**) Distribution of mean remapping scores $\bar{z}_{\Delta\tau}$ by unit, note the negative shift in the distribution of scores. (**e**) Scatter-plot between a session's task performance $p_{se}$ and the remapping scores $\bar{z}_{\Delta\tau}$ for recorded units. Size and color of dots scale with the x axis for illustration. Regression line in red with a $CI_{95\%}$ band. Kendall correlation score between behavior and remapping score shown. (**f**) Scatter-plot between $p_{se}$ and the mean remapping score across a sessions units $\bar{z}_{\Delta\tau\,\bar{s}e}$. Size of dots indicate number of co-recorded units, color codes correspond to different subjects. Regression line and corresponding $CI_{95\%}$ band shown in grey. (**g**) Like (**f**) but with neural population correlation, composed of the spatial rate maps for all recorded units in a session. (**h**) Correlation between $p_{se}$ and $\bar{z}_{\Delta\tau\,\bar{s}e}$ by subject, with bootstrapped standard deviation (B=500). $\bar{s}$ is the across subject mean.

The online version of this article includes the following figure supplement(s) for figure 4:

**Figure supplement 1.** Spatial rate maps of reward selective units.

**Figure supplement 2.** Spatial rate maps of direction selective units.

**Figure supplement 3.** Reward remapping vs behavior control analyses for isolated units.

**Figure supplement 4.** Reward remapping vs behavior control analyses for isolated units and MUA.

**Figure supplement 5.** Relationship between remap scores for cue and reward.

**Figure supplement 6.** Cue and reward rate coding vs correct/incorrect interaction.

**Figure supplement 7.** Correct vs incorrect rates during the transition between Outbound and Inbound (return) trajectories.

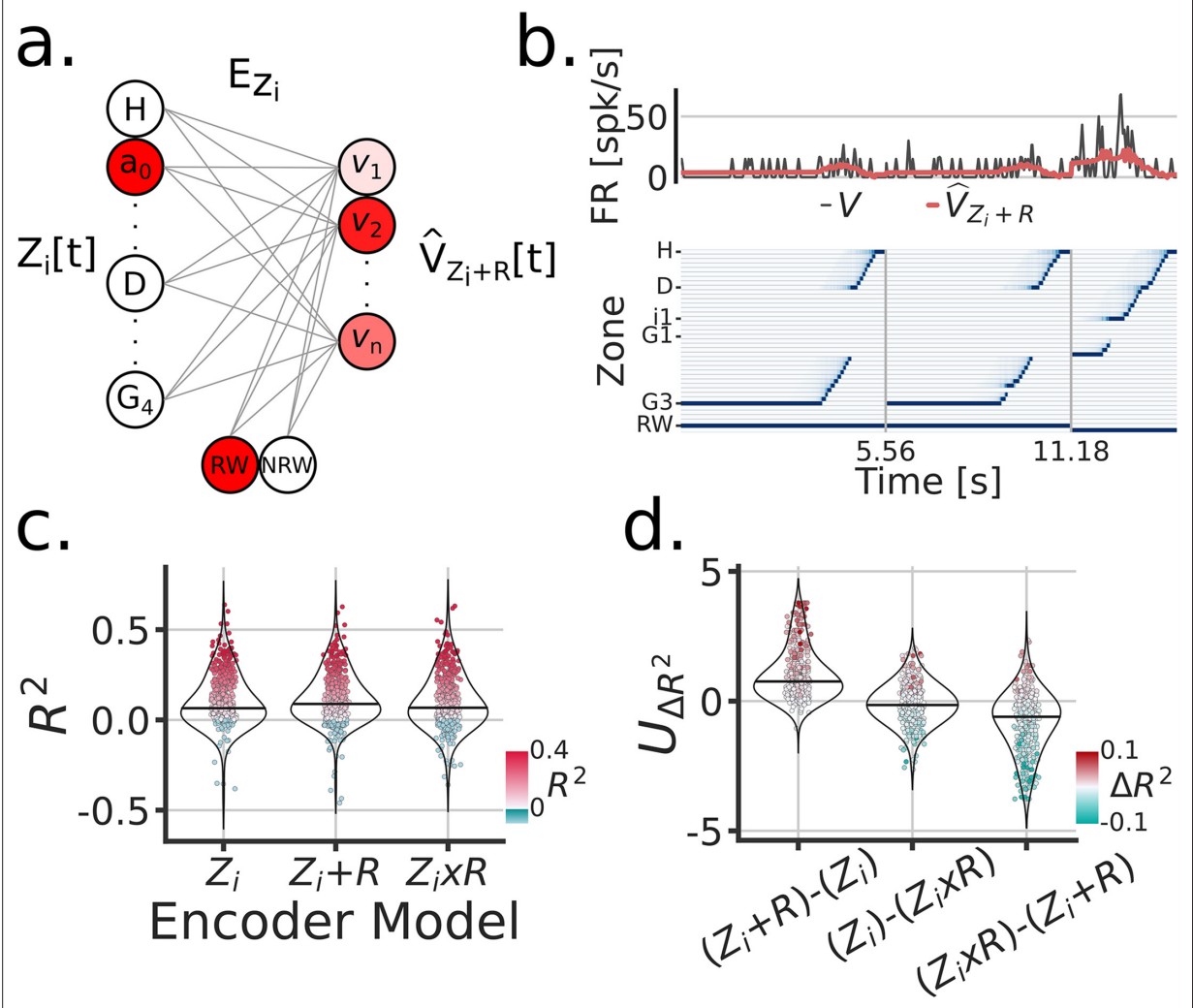

**Figure 5.** Encoding model of reward remapping. (**a**) Linear zone encoding model with reward $Z_i + R$, at a given sample time $t$, the current position of the animal and the reward identity is multiplied by learned weights to predict FR for each recorded unit $\hat{V}_{Z+R}$. (**b**) Example time window of $Z[t]$, the true FR in black $V[t]$ and the predicted FR in red $\hat{V}_{Z_i+R}[t]$. (**c**) Model comparison between three types of zone Encoding during inbound trajectories: $Z_i$, $Z_i + R$, $Z_i \times R$. Each dot is a unit, blue dots were negative $R^2$, red scales with $R^2$ and y-axis. (**d**) Model comparison scores. Y-axis is the Mann-Whitney Z transformed statistic for comparing the $R^2$ on test folds. Color-bar indicates the median difference in $R^2$ across test folds.

Furthermore, because rewarded trials are correct trials, we hypothesized that as with the cue, remapping scores ($\bar{z}_{\Delta\tau}$) should relate to task performance. The extent of reward related remapping correlated with performance accuracy for a session at the level of individual units, across co-recorded units, and individual subjects (*Figure 4e–h*, *Figure 4—figure supplements 3 and 4*). Statistical analyses of these results were indicative of a strong relationship between reward remapping and task performance: unit-level (Kendall's $\tau = -0.28$; LMEM: $\bar{z}_{\Delta\tau}$ slope $= -0.14$, $CI_{95\%} = (-0.18, -0.11)$, $LRT = \chi^2_1 = 84.9$, $p = 3.06 - 20$); across co-recorded units ($\tau = -0.43$; LMEM: slope $= -0.28$, $CI_{95\%} = (-0.38, -0.19)$, $LRT = \chi^2_1 = 30.54$, $p = 3.28e^{-8}$). Similar to cue induced remapping, the extent to which units remapped in response to reward receipt significantly correlated with behavioral accuracy in the task.

As with the cue, we modeled the firing rate activity of individual neurons through three linear encoding models: (1) a zone/position model $Z_i$, (2) a reward rate-remapping model $Z_i + R$, and (3) a reward global-remapping model $Z_i \times R$. Where the $i$ subscript indicates inbound trajectories, *Figure 5a and b*. Similar to the cue models, we found that the activity of most units were better fitted by a rate-remapping model (*Figure 5c, d and e*) (LMEM: $U_{\Delta_{R^2}}$ Model $LRT = \chi^2_3 = 856.9$, $p = 1.98e^{-146}$)

(*Figure 5d*). Thus these set of results indicate that parahippocampal neurons (as a population) exhibit increased firing rates during inbound trajectories that follow unrewarded trials.

Given the observations of spatial remapping for both cue and RW condition contingencies, we sought to understand the relationship between these results. First, we found that remapping scores were correlated (*Figure 4—figure supplement 5*) across individual units and at the population level. Second, because correct/incorrect trial contingencies largely contributed to how cue and RW spatial maps were constructed, we partitioned the data into units that showed correct versus incorrect coding on the basis of their patterns of activity across maze segments (Methods). The correct/incorrect groupings were formed independently for cue (outbound) and RW (inbound) spatial maps, and we sought to find how the correct/incorrect groupings overlapped across cue and RW, (*Figure 4—figure supplement 6*). With this approach, we found that incorrect coding units were more likely to overlap between cue and RW than correct coding units. However, remapping scores didn't differ between these unit groupings. Third, we conditioned the activity of correct and incorrect inbound trajectories based on the activity of correct and incorrect outbound trajectories (*Figure 4—figure supplement 7*). This analyses was motivated based on the structure of the trials: rewarded (correct) inbound trajectories followed correct outbound trajectories and that unrewarded (incorrect) inbound trajectories followed incorrect outbound trajectories. We found that incorrect >correct activity levels on outbound trajectories predicted incorrect >correct activity levels on inbound trajectories, suggesting that error signaling activity persisted through the inbound phase of the trial. Together, the remapping observations provide a quantification of this error or mismatch-like signal through different phases of the task and in the context of spatial representations.

## Units encoding head direction during open field foraging are more likely to remap

In addition to the Tree-Maze, single units were recorded from subjects while they foraged for randomly scattered food rewards in a rectangular open arena (open-field foraging, OF, *Figure 6a*). Removable panels forming the floor of the arena were placed on top of the Tree-Maze, thus keeping peripheral visual cues relatively constant between the two environments, with the main difference coming from the 12 cm in gained height for the OF environment. A model-based encoder approach was used to quantify the extent of navigational coding for each unit along the behavioral measures of speed, head-direction or position (*Figure 6*). The firing-rate predictions of these individual models was used to predict the firing rate of single units on test data in an aggregate model (*Figure 6a and b*). Across units, the aggregate model produced better results than individual models (LMEM: main effect of model $LRT_{R^2} = \chi^2_4 = 567.15$, $p = 1.33e^{-122}$, $LRT_{r_p} = \chi^2_4 = 878.84$, $p = 3.44^{-190}$) (*Figure 6c and d*). Units across the Tree-Maze and OF tasks were matched through a waveform matching algorithm *Figure 7a* (*Figure 7—figure supplement 1*). In total 217 units were matched across 76 OF and 82 Tree-Maze sessions *Figure 7b*. The set of model coefficients (speed, head-direction, position) for each matched unit were clustered into three groups (*Figure 7c–e*). Notably, the obtained unit cluster identified a main dimension of separation between clusters in the form of a coding trade-off between head-direction and speed (*Figure 7d and e*). That is, units that coded strongly for head-direction (cluster 0, blue, n=98) coded poorly for speed, while units that coded strongly for speed coded poorly for head-direction (cluster 2, green, n=59). For completeness, OF tuning score metrics were also computed for these clusters with results qualitatively matching the model coefficients (*Figure 7f*).

Next, we sought to relate how these clusters of units defined in the OF foraging task related to the results observed in the Tree-Maze. We observed that on this matched subset of neurons, significant remapping scores were still observed for both cue and RW contingencies, and that units in cluster 0 remapped more than the other clusters (LMEM: cluster x remap interaction $LRT = \chi^2_2 = 6.87$, $p = 0.032$; cluster $LRT = \chi^2_2 = 1.5$, $p = 0.47$), (*Figure 7g*). Similarly, Tree-Maze encoder rate/global remapping scores were higher for cluster 0 (LMEM: cluster $LRT = \chi^2_2 = 13.89$, $p = 9.62e^{-4}$) (*Figure 7h*). Additionally, we explored the extent to which these clusters related to other aspects of coding in the Tree-Maze like correct/incorrect coding and cue/RW coding by segment (*Figure 7—figure supplements 2 and 3*). We found that these functionally defined units had some overlap with different aspect of the maze, suggesting a small degree of heterogeneity in coding. However, most units in the clusters did not overlap with cue/RW or correct/incorrect coding. Taken together, units in cluster 0, that corresponded to stronger head-direction coding, were more prone to remap for both cue and RW than

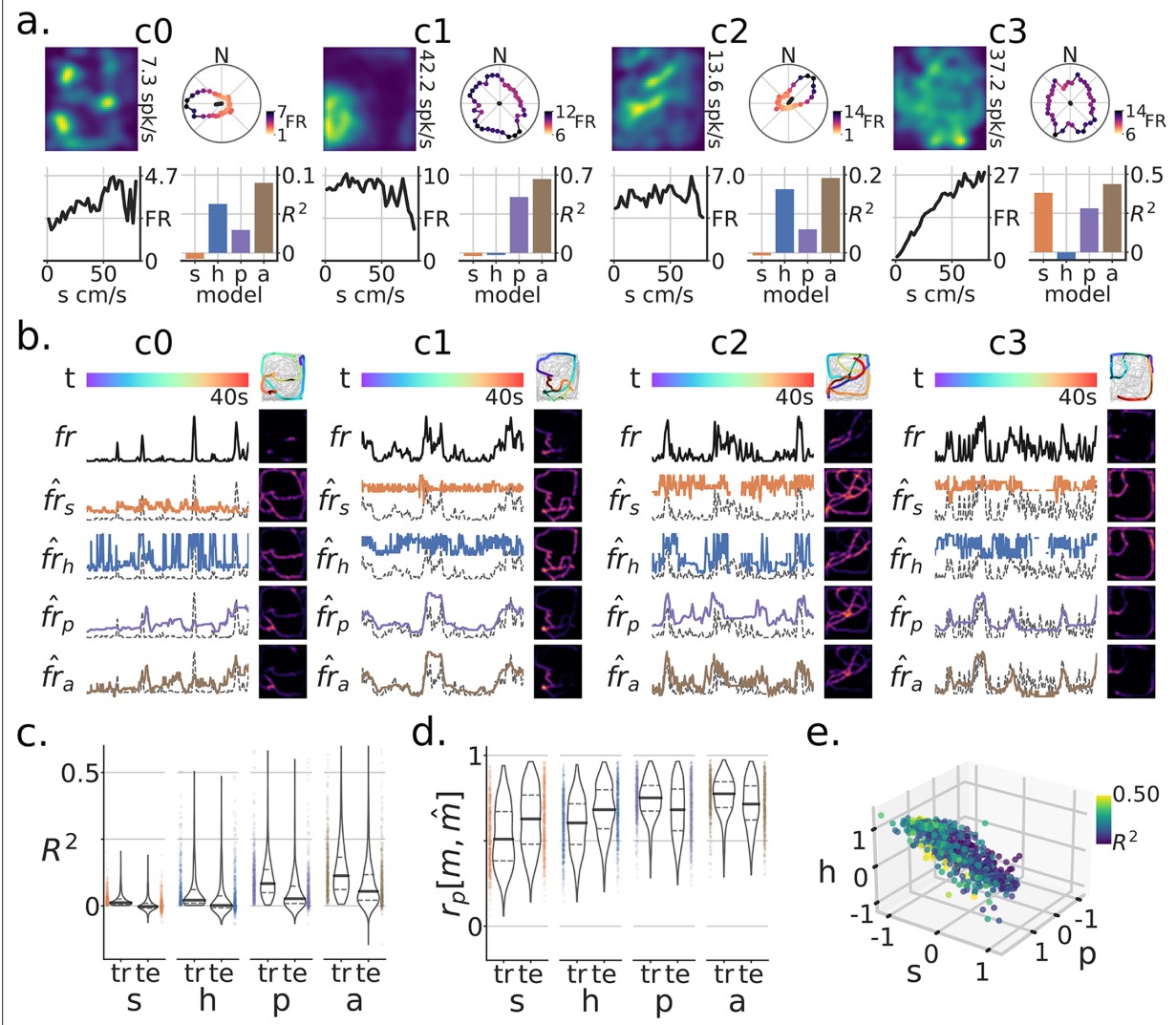

**Figure 6.** Modeling navigational/spatial variables in neural coding during open-field foraging. (**a**) Neural responses of four example units for subjects foraging an open-field (OF) arena [1.3m x 1.5m]. For each unit sub-panel (**c0–c3**): top-left, firing-rate map (number is the peak FR); top-right, head-direction tuning curve, color indicates FR magnitude by angle; bottom-left, speed tuning curve (s=speed); bottom-right, model-based variance explained on test data ($R^2$) by variable (h=heading-direction, P=position, a=aggregate model). (**b**) Model-based responses by variable for a selected test time-window for units c0-c3. Each row corresponds to a different model prediction ($\hat{fr}$), the true firing-rate (fr) for that unit, or at the top the color-coded time-window (**t**). Top-right, the data on which the model was trained is in grey and super-imposed is the test-window color-coded by time and with firing rate magnitude in increasing dot-size. Other heat-maps are the resulting firing-rate maps generated for the test trajectory. Model predicted rates are shown with colors matching (**a**), with the background dotted line being the true (fr) (**a**). (**c**) Population level ($R^2$) for train (tr) and test (te) sets. (**d**) Population level firing-rate map Pearson correlation $r_p$ between true $m$ and predicted maps $\hat{m}$. Note that for both metrics, the aggregate model and the position model produced the best results. (**e**) Relationship between coefficients on the aggregate model by unit. The color corresponds to the model's training set $R^2$.

units in cluster 1 or 2. Hence, we leveraged Open-Field recordings to identify head-direction coding units as showing changes of firing patterns in the Tree-Maze, with those changes being reflective of task performance.

## Discussion

During Open-Field foraging, individual MEC, PrS and PaS neurons encode an animal's position, orientation and speed but the firing patterns of neurons in these parahippocampal regions during goal-directed navigation has been less well characterized. Here, we examined the firing patterns of

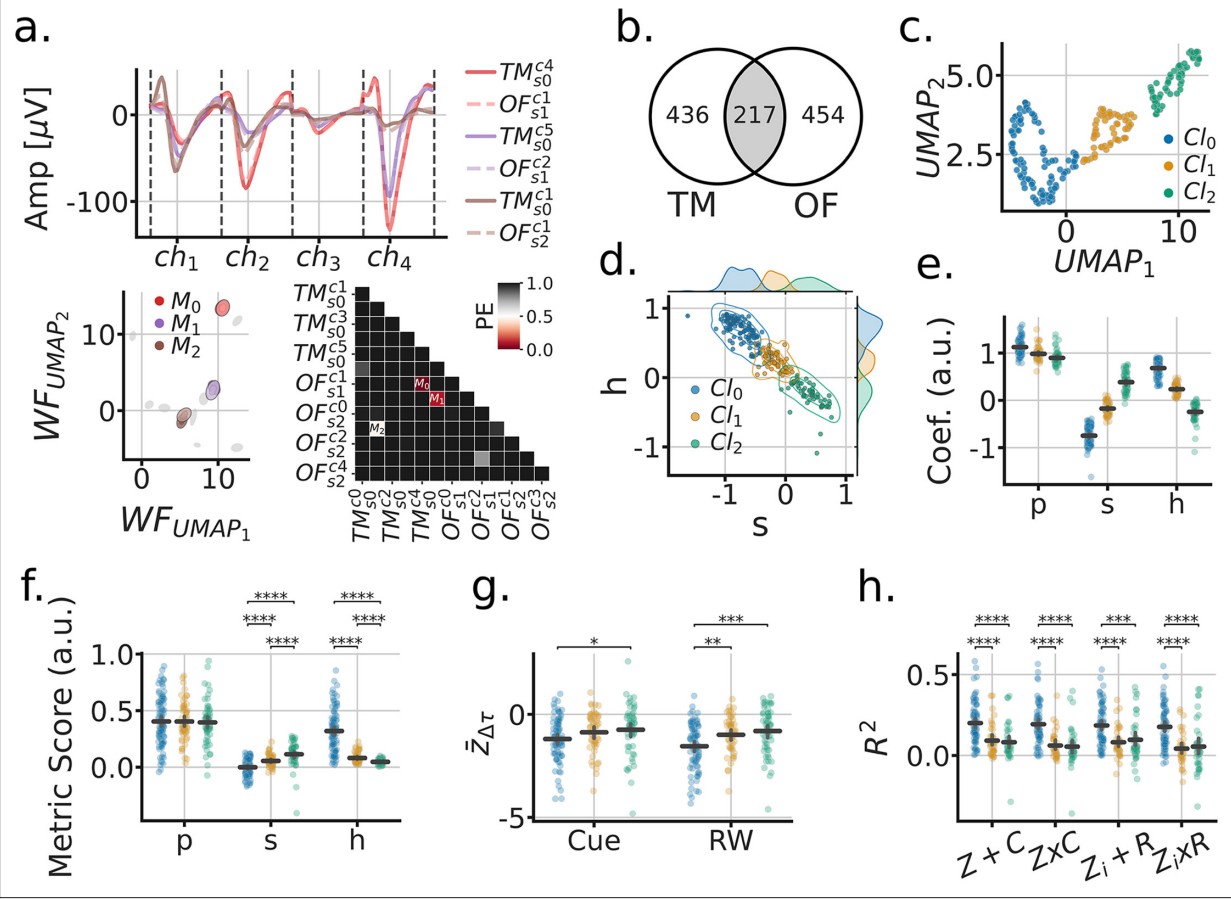

**Figure 7.** Matched units across tasks reveal that head-direction coding units remap the strongest. (**a**) Procedure for matching units across tasks (OF-open field, TM-tree maze). Top, 6 tetrode waveforms color-coded by matched units; bottom-left, fitted Gaussians to the dimensionally reduced unit waveforms for that tetrode across matched depth sessions (grey, unmatched units); bottom-right, symmetric confusion error matrix across units, threshold for matching $PE <= 0.5$. (**b**) Venn diagram of matched units across tasks. (**c**) UMAP clustering of the aggregate model coefficients for matched units. (**d**) Head-direction (**h**) vs speed (**s**) coefficients formed a clustering subspace. (**e**) Model coefficients of aggregate model used for finding clusters ($P$=position). Horizontal lines are the population mean, and error bars are the mean's 95% CI. Statistics not performed, as the clusters were fitted from these parameters. (**f**) Tuning metric scores ($P$=split half rate-map correlation; s=speed-score; h=resultant-vector length score). Paired statistics through Mann-Whitney U tests (*= $p < 0.05$, **= $p < 0.01$, ***= $p < 0.001$, ****= $p < 1e^{-4}$) (**g**) Remap scores by OF cluster. (**h**) TM model scores by OF cluster.

The online version of this article includes the following figure supplement(s) for figure 7:

**Figure supplement 1.** Demonstration of waveform matching algorithm.

**Figure supplement 2.** Cue and reward rate coding vs correct/incorrect interaction and cluster identity.

**Figure supplement 3.** Differences in activity rates by cluster and condition.

hundreds of parahippocampal neurons as rats navigated a visually-guided goal directed Tree-Maze task. Rats successfully learned to use a visual cue to navigate the left or right branch of a multi-decision point linear maze to receive reward. We found that many neurons exhibited higher firing rates after incorrect left or right branch choices, which led to remapping of spatial firing rate maps. Further, the degree of remapping correlated with behavior, such that higher remapping scores corresponded to better task performance. Using encoder models, we then demonstrated that rate remapping, over global remapping, was sufficient to explain the observed changes in spatial firing rate maps. Finally, by using the navigational coefficients of encoder models fit to Open-Field foraging data, we further revealed that head direction coding cells showed the largest magnitude of remapping on the Tree-Maze. Conjunctively, these results demonstrate that parahippocampal representations of space and navigation, under goal-directed experimental conditions, project a flexible code reflective of behavior.

Neuronal encoder models serve as an important tool for identifying single-unit activity coding during open-field foraging that avoid imposing a prior on the shape of tuning curves or the generation

of coding indices (e.g. tuning curve scores) (*Hardcastle et al., 2017b*). Moreover, these models allow for the quantification of mixed selectivity in the coding for spatial/navigational variables, providing a more nuanced understanding of the possible variables a neural population encodes. Here, we fit linear encoder models that used speed, heading-direction and position during open-field foraging to predict each unit's firing rate activity. Consistent with prior work, we find strong position coding, that is often complementary with coding for head-direction or speed (*Hafting et al., 2005*; *Hardcastle et al., 2017b*; *Diehl et al., 2017*; *Rowland et al., 2016*). Although there are differences in the proportion of cells that encode position, head direction and speed between MEC, PrS and PaS (e.g. PrS and PaS have higher proportions of head-direction tuned cells), these regions share a common substrate of spatial/navigational coding (position, head-direction, speed) (*Boccara et al., 2010*; *Spalla et al., 2022*; *Tang et al., 2016*). We further found a relationship between navigation variables after clustering that revealed an axis of separation along the head-direction and speed coefficients, such that these variables were negatively related. This anti-correlation between speed and head-direction coding replicated recent observations across parahippocampal regions (MEC, PaS, and PrS) (*Spalla et al., 2022*). Thus, as a function of speed, the head-direction signal monotonically decreases at faster speeds, consistent with the observation that subjects are less likely to turn when they run fast. Thus, one possibility is that the head-direction signal in the parahippocampal region reflects a behavioral state related to navigational choice or the lack of commitment to a particular navigational route.

Prior works point to the MEC as a key node supporting spatial decisions, as it's inactivation led to performance deficits in visual-scene based spatial decisions (*Yoo and Lee, 2017*), and subsets of MEC neurons encode the near future position of the animal (<1 second in the future; *Kropff et al., 2015*; *Campbell et al., 2021*). Nonetheless, in terms of navigation behavior, parahippocampal neural activity after incorrect decisions has been understudied. Depending on the task, there are often very few error trials or multiple confounds that prevent substantial quantification of errors (e.g. loss of attention or satiation). Error or mismatch signals conform to a fundamental computation that informs immediate behavior, learning and neural plasticity (*Gershman and Uchida, 2019*; *Sosa and Giocomo, 2021*). Our task structure enabled us to more rigorously consider error trials and the activity of MEC, PaS, and PrS neurons before and after such trials. Given that many units exhibited higher firing rates on incorrect trials, leading to rate remapping, we hypothesized that the magnitude of remapping should correlate with behavioral performance. Indeed, we observed strong correlations between neural remapping and behavioral performance across multiple levels of analyses: individual units, across units, population correlations, individual subject level analyses and a neural population decoder model that mimicked the subject's behavior (*Figures 2–4*), as well as in control analyses that included multi-unit activity or the removal of periods of reward and/or immobility (*Figure 2—figure supplements 6 and 7*, *Figure 4—figure supplements 3 and 4*). The robustness of this observation suggests that cortical regions encode error, expanding the possible brain regions involved in error signalling beyond downstream regions, like the basal ganglia, that prior work has shown carry error signals (*Gershman and Uchida, 2019*; *Sosa and Giocomo, 2021*).

Additional support for the idea of a mismatch signal in the parahippocampus comes from prior works that observed changes in the hippocampus related to correct or incorrect decisions. For example, hippocampal place-cells showed more efficient representations of sequences during correct trials than during incorrect (*Zheng et al., 2021*), and non-place cells show higher rates during correct versus incorrect trials (*Zhang et al., 2022*). Moreover, in the T-maze, the extent of discriminability for upcoming left or right trajectories in the firing rate of hippocampal cells active on the stem of the T-maze (i.e. splitter cells) correlated with behavioral performance (*Kinsky et al., 2020*). Of note however, we did not observe a significant population of 'splitter cells' in the central stem of the maze, possibly because the Tree-Maze differs from previous T- or Y- Maze in that there is a reward well at the decision point (*O'Neill et al., 2017*; *Frank et al., 2000*; *Lipton et al., 2007*). Even so, in our task, after training, subjects were more likely to be running on the left branch of the maze when the green visual cue (LC) was present, but after an incorrect decision, they were running on the left branch of the maze when the purple cue (RC) was present. These incorrect spatial decisions led to increase firing rates on the majority of units, and persisted on the inbound trajectories (quantified by rewarded vs unrewarded, *Figure 4—figure supplement 3*). Notably, units coding for correct decisions on outbound trajectories rarely overlapped with units coding for correct decisions on inbound trajectories, while neurons coding for incorrect decisions often overlapped in their coding for inbound and outbound

trajectories (*Figure 4—figure supplement 2*). These observations are generally consistent with the idea that a mismatch signal could be transmitted by MEC, PaS, and PrS neurons via a change in firing rate. However, it remains an open question as to where this type of mismatch signal originates such that it is broadly reflected across parahippocampal regions.

A growing set of studies have started to show that spatial and navigational codes in the parahippocampal region are more flexible than previously thought. Observations of spatial remapping in the MEC have been seen after significant changes to the environment in an open-field arena (*Butler et al., 2019*; *Marozzi et al., 2015*), provoking a coordinated re-organization of grid-cells and seemingly random re-organization of other spatially selective cells. Remapping was also observed to be more likely at low running speeds (*Low et al., 2021*), switching between stable spatial representations. These observations would generally be considered global remapping, as the activity fields in space change position. Other studies, however, have shown changes in firing rate and small shifts in fields near rewarded locations during foraging (*Butler et al., 2019*; *Boccara et al., 2019*). Our data, including deep MEC, points to changes in firing rate activity on the same spatial location (rate remapping) being sufficient to explain the differences in firing rate observed between conditions (both for cue and reward). The rate remapping result is again consistent with the idea of a mismatch or error signal acting as a gain on individual neurons. Our results demonstrate that through the use of a dynamic environment we increased the behavioral demands on the animal, providing a behavioral assay to contrast parahippocampal neural activity. Further, by recording from the well characterized parahippocampal navigational circuit, we were able show that there are additional dimensions of representation or computations only elucidated through the use complex behaviors.

## Methods
### Subjects
All techniques were approved by the Institutional Animal Care and Use Committee at Stanford University (IACUC) under protocol (#27694). The subjects in this study were male Long Evans rats (Charles River Laboratories, n=5) that were housed with liter-mates and kept on a 12 h light-dark cycle. Rats were moved to solitary housing and handling began when rats were 12–16 weeks old (400–500 g). After a one week handling period, animals were introduced to the Tree-Maze environment, and behavioral training began. After animals were acclimated to the maze and we obtained their base-weight, a dietary food restriction was introduced. The animal's weight was monitored daily, and kept at 90% of their base-weight throughout behavioral training by controlling their daily food intake. The dietary restriction motivated the animals, while still keeping them at a healthy weight. Food restriction was paused at least 3 days prior to surgery, and for a week post-surgery while animals recovered. During this recovery period, the animal's health was monitored daily for signs of infection, or other behavioral indicators of a lack of recovery.

### Surgery
Surgeries were conducted in a designated laboratory space using aseptic technique. All rats used in the project were implanted with a micro-drive and ground wire for electrophysiological recordings (Halo-18 Neuralynx, Inc). During surgery, the rat's head was held in a stereotaxic carrier and heat was provided by a self-regulating heating pad. The entire procedure took approximately 5 hr. Anesthesia was isoflurane with buprenorphine as a pre-anesthetic sedation (0.05 mg/kg subcutaneous). Ophthalmic ointment was applied to prevent eye drying. Subcutaneous fluids (0.9% NaCl) were administered during the procedure (5 mg/kg/hr). For the surgery, the top of the skull was first clipped and prepped, and then a midline incision was made to expose the top of the skull. A hand drill was used to secure several orthopedic screws to the skull (one serves as an electrical ground for data acquisition). After the screws were secured, a ~1 mm diameter wide hole was drilled directly above the medial entorhinal cortex. The location of the hole was determined using a stereotaxic atlas (coordinates from Bregma ~–8.5 AP, 4.5 ML), and landmarks on the skull (hole was tangential to the side skull ridge and over the extended lambda skull crevice). During drilling, the skull and holes were kept from excessive drying by applying sterile saline, and drilled only deep enough to expose the dura. Using a sterile dura cutting tool, a small incision was made and the recording electrodes were slowly lowered into the brain (to a depth of 1000 microns). Surgical adhesive (Kwik-Sil) was used to fill in the space between

the implant and the skull hole. With the implant in-place, the ground wire was connected to one of the ground screws, and carefully sealed with silver conductive ink. After curing, several layers of adhesive cement (C&B Metabond) was applied to cover the screws and implant. Finally, dental acrylic cement was used to adhere the metabond layers to the rest of the implant. Sterile sutures were applied to secure the skin around the implant. Following the surgery, the rats recovered on a circulating warm water blanket and were checked every 15 min until mobile.

## Histology

Upon completion of the final recording session, rats were anesthetized with 2–4% of isoflurane in $O^2$ and euthanized using an overdose of pentobarbitol followed by transcardial perfusion with saline and 10% formalin. After the brain was stored in formalin for at least 24 hr, the hemisphere containing the implant was sliced into 25 µm sagittal sections using a cryostat microtome. Sections were mounted on slides and Nissl-stained with cresyl violet. Microdrive damage to the tissue was extensive and exact tetrode locations were difficult to pinpoint across the regions of interest (MEC, PaS, PrS), mostly due to the geometry of the recording bundle. Rough coordinates of tetrodes ranged from 1 mm from the cortical surface to 4 mm (Dorso-Ventral), 7.5–9 mm from Bregma, and 3.6–4.6 mm to the right of the medial fissure (Medio-Lateral). Slices for the five animals reported can be seen in *Figure 1—figure supplement 2*.

## Recordings

Electrophysiological signals were recorded using a Digital Lynx 4 Sx data acquisition system with up to 64 channels (Neuralynx, Inc). A head-stage amplifier was attached to the subjects implant before each recording session, and the connecting light-weight cable was routed through a power commutator (Neuralynx Inc), which then connects to the acquisition system. Data was collected at 32 kHz, and digitized at 16 bits. We used the Cheetah interface (Neuralynx, Inc) for data collection and monitoring. All of the events from the environment were sent via a micro-controller to the Digital Lynx TTL input board. One of the cameras synced with the Cheetah interface, and was set up to save color channels (at a user defined luminance threshold) for the detection of the subject's location (LEDs for tracking are located on the head-stage amplifier). The processed video data only saved x, y positions and head angle. A second high-resolution/high-rate color camera worked independently to record the animals' behavior in the maze (Basler Inc).

## Behavioral Rig

### Housing frame

A custom designed aluminum frame (80/20 Inc) housed the behavioral apparatus, two high resolution cameras and commutator (dimensions: 1.3 m width, 1.5 m length, 2.5 m height; *Figure 1—figure supplement 1*). The frame was connected to a ground port of the recording system and the behavioral micro-controller device. A series of supports beams were placed 1.35 m above ground to hold the large 'floor' panels that make the open-field setup. Black polycarbonate panels (0.5 m height) around the frame were placed at 1.37 m, which allow the floor panels to slide into the aluminum frame for open-field sessions. Signal cables and liquid reward tubing are routed through the indentures of the frame, where possible.

### Tree-Maze

We adapted the Tree-Maze behavioral apparatus from *Joo et al., 2021*, Figure 1a (see also *Ainge et al., 2007* for a similarly structure maze). The overall dimensions of the maze were 1.4m x 1.2m, and it was suspended 1 meter above the ground in the Behavioral Rig aluminum frame. The maze featured six reward wells, each equipped with infra-red detectors for nose-poke detection and a white LED light that indicated the presence of reward. The reward wells were custom designed with luer-lock ports, space for IR detectors and the LED. The wells were 3D printed with a ONYX carbon plastic material, which provided the required durability. The luer-lock ports are used to connect to food-grade PVC tubes that transport the liquid reward. These wells also have an ethernet port that connects them to the electrical controls (LED, IR detection signal, power and ground). The control port and reward port are located below the maze, such that the animal has no visibility or access. The cue panel is a 16x16 RGB LED programmable array, placed on the back wall of the apparatus and

40 cm above the maze. Liquid milk rewards (80% evaporated milk / 20% H20 +20 grams of sucrose) are delivered via a custom designed pump system. The 6-pump system consists of linear actuators (3inch-12Volt actuators, Progressive Automations Inc) operating on milk filled syringes. This system of pumps is enabled via an array of relays and custom circuit boards with switches, allowing both manual and programmatic control of the pumps. A custom designed housing box encloses the electronics and relays, with external attachment features to hold the pumps, syringes, switches and power input.

Low-level control of the environment (LEDs, sensors, reward delivery, cue, pulse event outputs;+5 V binary signals) is achieved through an Arduino Mega micro-controller, with a custom designed input/output interface. Each well, the cue, and the pump system are connected to this interface via ethernet cables (RJ45). A Python custom-coded state machine framework is used to control the Arduino via USB. The PC-operated custom Python code provided a command line interface to monitor and control the environment parameters online (e.g. amount of reward for a well, well LEDs, cue, switch probability). Events are saved on the computer and are sent as TTL inputs to the electrophysiology recording device. Environmental control is enabled through the state-machine-like software architecture, in which states are defined by which well is enabled to give out reward. This platform allows for quick customization of environment parameters and task structure. Designs and software are available on https://github.com/alexgonzl/TreeMaze (copy archived at *Gonzalez, 2019*).

## Open-Field

Open-Field foraging experiments can be performed on the same Behavioral Rig, by sliding 'floor panels' on the housing frame. The resulting field sits atop 20 cm above the Tree-Maze, with all other peripheral cues remaining the same, including the LED cue panel as an in-environment cue landmark. The dimensions of field are 1.3m x 1.5m.

## Behavioral training

After subjects are acclimatized to human handling and receiving solid food reward (Cheerios), they are placed in the Tree-Maze environment. In initial sessions, subjects simply explore the environment that contain Cheerios at the reward wells, such that they associate these locations with reward. Once the location-reward association is established, the food restriction protocol begins (see the Subjects section). The training protocol then follows these stages: (0) introduction to milk-reward, (1) LED light - reward conditioning, (2) task-route training, (3) cue L/R navigation, and (4) reward wells LED off. In Stage 0 (~1 session) the solid rewards are replaced with milk rewards. Stage 1 takes about 2 sessions to master, and in this stage the milk rewards are only provided after triggering (nose poke Infra-Red beam-break >10ms) active wells which have an ON LED light. After a detected beam-break on an active well, the milk reward is delivered and the LED light is turned OFF. A log of number of activated wells was kept to track progress. Stage 2 (~5 sessions to a 20 trial criteria) introduces the basic trial structure of the task by having reward sequences in the following pattern: (1) Home-well (H) → (2) Decision-well (D) → (3) Goal-well (G) → (4) H. In this training stage, any one of the Goal-wells (G1-G4) can be rewarded at the third point of the sequence; and once a goal is triggered the rewards on the others are no longer available. The subject then needs to return to H and trigger it to commence a new trial. Once subjects can perform more than 20 trials, they advance to the next stage. Stage 3 (~32 sessions to 75% behavioral accuracy and 80 trials criteria) is the final stage of training, it has two parts, and once subjects reach the performance and number of trials criterion they are candidates for surgery (*Figure 1—figure supplement 2*). In Stage 3 a, in addition to the sequence introduced in Stage 2, subjects now see a cue after triggering the HW at the beginning of a trial. The cue can be green pulsating at 7 Hz indicating a left turn, or purple pulsating at 3 Hz indicating a right turn. These turns indicate where the animal needs to turn after D to receive further reward at a goal. For a turn right cue (purple), the subject will need to turn right after the triggering D, and triggering one of G1 or G2. However, only one goal (experimentally chosen at random) on the right branch will have reward but both have LED lights on to indicate possible reward. Similarly for a turn left cue, and G3-4. Cue color wavelengths, green and purple, were selected to differentially excite the rat's dichromatic vision system (*Jacobs et al., 2001*). In Stage 3b, the LED lights on the goals never light up, and after D the subject only has the ongoing cue signal to make a turn decision. Notably, the H and D retain their LED lights

ON at the appropriate times in the trial sequence. Well trained rats often exhibited stereotyped trajectory behaviors, such that they visited goal wells within each branch in the same order.

An error/incorrect/unrewarded trial is one in which the subject triggers a Goal well that is not in the branch associated with that trial's Cue. If the subject first navigates to the incorrect branch, doesn't trigger a Goal well and then returns to the correct branch to trigger a Goal well, this is marked as a correct trial. However, these trials are excluded from analyses, as there very few of them and the 'change of mind' aspect makes these trials fundamentally different than the simple 'correct' and 'incorrect' labels. On the inbound trajectories, rewarded and unrewarded trial labels refer to the results of the immediately preceding outbound trial, correct and incorrect trials, respectively.

## Data processing

### Electrophysiolgy pre-processing

Individual broadband traces for each channel are characterized includes quantification of clipped segments, power spectrum peaks, and amplitude histogram. This information is later used to determine if channels should be excluded from analyses. Each trace is digitally filtered using filter banks of IIR Second-Order-Sections (high-pass 2 Hz, low-pass 5 kHz, notches at 60 and 120 Hz).

### Unit clustering

The Kilosort2 algorithm (*Pachitariu et al., 2016*), open-source python packages (SpikeInterface *Buccino et al., 2020*), and custom python scripts were cell clustering for each recorded session. For Kilosort, mostly default values were used, with modifications appropriate for tetrode recordings (probe geometry, no common referencing and no data whitening). Manual curation was used to classify the resulting clusters from Kilosort as cells (units), multi-unit (MUA), or noise. Cluster statistics and heuristics used for classification were inter-spike interval violations, signal-to-noise ratio, waveform shape, and cross-correlogram relationship with other clusters within the session.

### Across session unit matching

In order to match unit across sessions the following algorithm was used. First, we matched tetrodes across sessions according to their depth for each subject. That is, this algorithm only tries to match units across sessions for which the corresponding tetrodes was at the same depth for those sessions. Second, a random sampling of 1000 spike waveforms for each unit across the matched sessions are pooled into a single data matrix $W$. A single spike waveform has a total of 32 samples (centered around the peak) for each tetrode channel for a total of 128 features. The size of the matrix is then 1000 times the number of units by 128. Each row of $W$ corresponds to a unit's waveform, denoted as $u_{k,i}$, where $k$ is the unit number and $i$ corresponds to the ith spike of unit $k$. Second, $W$ is then subjected to UMAP for unsupervised dimensionality reduction into 2 dimensions $W_{\mathrm{UMAP}_2}$. Third, for each unit $u_k$, robust mean and covariance estimates are obtained via (Minimum Covariance Determinant-MCD, *Rousseeuw and Driessen, 1999*), such that the 2-d UMAP representation of wave-forms for unit $k$ can be described with a compact 2d Gaussian $f_k \sim \mathcal{N}(\mu_k, \Sigma_k)$. We then create a distance metric based on the expected probability of cluster mis-classification error:

$$f_k(x) \quad \sim \mathcal{N}(\mu_k, \Sigma_k)$$

$$= \frac{1}{2\pi|\Sigma_k|} e^{-\frac{1}{2}(x-\mu_k)^T \Sigma_k^{-1}(x-\mu_k)} \tag{1}$$

$$E_{kk} = E[f_k(x)|x \in u_k] \tag{2}$$

$$E_{kp} = E[f_k(x)|x \in u_p] \tag{3}$$

$$E_{pk} = E[f_p(x)|x \in u_k] \tag{4}$$

$$ne_{kp} = 1 - \frac{E_{kp}}{E_{kk}} \tag{5}$$

$$d_{kp} = d_{pk} \quad = \frac{1}{2}(ne_{kp} + ne_{pk})$$

$$= 1 - \frac{1}{2}\frac{E_{pp}E_{kp} + E_{pk}E_{kk}}{E_{kk}E_{pp}} \tag{6}$$

Note that the $x$ in the above equations represents the 2-d UMAP waveform representation of one spike. The value of $E_{kk}$, *Equation 2*, is the expected value of $f_k$ evaluated on all the entries of $W_{\mathrm{UMAP}_2}$ that are from unit $u_k$, and serves as a normalizing factor the computations of normalized error 5. The value of $E_{kp}$ is the expected value of evaluating spikes from unit $u_p$ in $f_k$, and this quantity represents the unnormalized distance of unit $u_p$ to the parameterized representation of unit $u_k$. The final step uses averages the normalized distances between $u_p$ and $u_k$ to obtain $d_{kp}$, the metric used in matching units across sessions, *Equation 6*. An important point is that the 2D representation provided by UMAP is not deterministic and relatively fragile to algorithmic parameters. Nonetheless, our procedure was robust for a large range of parameters and replicated using t-SNE instead of UMAP. We considered two additional metrics to evaluate distance metrics Hellinger distance $H^2$ and Kullback-Leibler divergence $D_{KL}$ (*Figure 7—figure supplement 1*). Equations provided below for reference in the context of multivariate Gaussians.

$$H^2(f_k, f_p) = 1 - \sqrt{2 \frac{|\Sigma_k|^{1/2} |\Sigma_p|^{1/2}}{|\Sigma_k + \Sigma_p|} \exp{-\frac{1}{4}(\mu_k - \mu_p)^T (\Sigma_k + \Sigma_p)^{-1}(\mu_k - \mu_p)}}$$

$$D_{KL}(f_k \| f_p) = \frac{1}{2}\left(\mathrm{tr}(\Sigma_k^{-1}\Sigma_p) + (\mu_k - \mu_p)^T \Sigma_k^{-1}(\mu_k - \mu_p) - 2 + \ln\left(\frac{|\Sigma_k|}{|\Sigma_p|}\right)\right)$$

In this context, the mis-classification error metric $d$ had some appealing properties in comparison to $H^2$ and $D_{KL}$. First, $d$ it is bounded from 0 to 1, with 1 indicating that the clusters do not overlap, while zero indicating that they perfectly overlap, and 0.5 indicating that both clusters are equally likely. Having a bounded metrics allowed for establishing a threshold that was consistent across multiple waveform scaling scenarios. Second, it is symmetric, $d_{kp} = d_{pk}$, a desired property in determining if the units are the same across multiple sessions. Note that $H^2$ is both bounded and symmetric. Third, it has an intuitive interpretation as the distance between unit waveforms (or the parameterized representations) with data points being evaluated and not depending completely on the Gaussian parameterization. In contrast, both $H^2$ and $D_{KL}$ will only be as accurate as the Gaussian parameterization of the unit cluster is (as computed through the minimum covariance determinant algorithm).

## Remapping
### Balanced trial resampling
An important component of the analyses that compared across conditions was obtaining a full and balanced coverage of the Tree-Maze environment. For example, for a session with 100 trials (40 LC / 60 RC) the actual coverage by cue condition will depend on the performance of the subject. If the subject performed at 50% for RC trials, the RC trial set is composed of 30 trials in which the subject navigated left after the Decision-well, and 30 trials navigated right. On average subjects perform at 75%, and on this example that means 45 trials in which the subject navigates to the right, and 15 in which the subject navigates left. Our procedure simply sampled an equal number of Correct and Incorrect trials per cue condition with replacement. This procedure was repeated 100 times for all statistical results reported, or 50 for plotting purposes. For generating the appropriate null distributions, even and odd trial sets were each balanced re-sampled by the number of LC and RC trials. Specifically, an equal number of LC and RC trials were sampled for each even and odd condition. The cue condition was defined by comparing Correct and Incorrect trials, and the balancing condition was cue. Sessions that do not meet a minimum of five trials in the nested conditions were excluded from these analyses.

### Firing rate maps
Traditionally firing rate maps are generated by pooling all samples and the corresponding spikes of a single unit to generate a single firing rate map. Because of the trial structure of employed in the Tree-Maze, we instead obtained trial-wise firing rates for each segment in the maze (including linear sub-segments n=39; *Figure 2b*). This approach then allowed the selection of trials to generate mean firing rate maps across trials. In the presence of possible outlying trials, this approach can generate a more robust estimate of the true representation of the environment. Outbound trial

trajectories are those in which the start is at when the subject receives the Home-well start trial reward (triggering the cue) and end at the last Goal-well detection. Inbound trial trajectories are those that start after the last Goal-well detection and end when the subject receives the reward to commence the next trial.

## Remapping scores

Remapping scores have two main components, a Test correlation score and a null correlation score. Each correlation score is the result of a Kendall correlation $\tau$ between mean firing rate maps for a given trial set. Thus, for a single remapping score there are four set of trials each used to generate firing rate maps, and two correlation scores $\tau_{Test}$ and $\tau_{Null}$. The use of Kendall ranked correlation provided robust measure of correlation. Zones with no samples in a given trial set are not included in subsequent calculations. Kendall correlations are first converted to a Pearson equivalent score: $r = \sin \frac{\pi}{2} \tau$, Fisher transformed $z = \arctan r$, and scores between the test measure and null measure are then compared with the following formula for comparing Fisher transformed correlations $z_{\Delta\tau} = \frac{z_1 - z_2}{((n_1-3)^{-1} + (n_1-3)^{-1})^{\frac{1}{2}}}$. This procedure was performed for each bootstrapped instance and for each unit, and averaged to obtain a final remapping score: $\bar{z}_{\Delta\tau}$.

## Statistics

Analyses of firing rate differences by condition were quantified using the Mann-Whitney U statistic (subsequently transformed to a Z statistic, $U_Z$). As this is a ranking statistic, it is more robust against trial outliers in firing-rate magnitude and to low trial numbers. Formal statistical analyses leveraged the power of Linear Mixed Effects Models (LMEM) to account for within subject variance, and repeated measures (units) in each session. This feature was of particular importance as there were different number of sessions and units by subject. For firing-rate analyses across units the condition difference statistic $U_Z$ was modeled as a function of maze segment. For remapping analyses, the remapping score $\bar{z}_{\Delta\tau}$ to explain a sessions performance $p_{se}$, with subject and task versions as random effects. Coefficient estimates $\beta_{\bar{z}_{\Delta\tau}}$ (slope of the linear model) were reported along 95% confidence intervals. In addition, a Log Likelihood Ratio Test (LRT) was performed against a null LMEM that excludes $\bar{z}_{\Delta\tau}$ as an explanatory variable, with significance testing using a $\chi^2$ distribution. The LMEM approach was employed in all reported statistical analyses. These analyses were performed using the Stats-Models Python package and custom python scripts.

## Tree-maze encoder/decoder

### Encoder

Data (behavioral/experimental variables and neuronal firing rate) were binned in 20ms windows, and each cross-validation (n folds = 5) training set contained an equal number of LC and RC trials, with each training sample weighted by the corresponding proportion of correct or incorrect trials. This cross-validation approach attempts to fully represent the space and task in a balanced manner. Position on the maze was modeled as a feature vector with a value of 1 on the feature location that corresponds to the subject's location at a given time sample (# of possible positions = $N_Z = 39$). The values on the rest of the feature vector were zero for the base model (one-hot vector). We also employed modified versions of this feature vector with an inverse lag decay function ($\left|\frac{1}{lag}\right|$) that modeled previous or future positions relative to the position at the given sample. These feature vectors were then used to train a Least-Squares Regression model to predict the firing rate of individual units on each 20ms sample bin. Three lag encoding models were used to fit the data: $F_{Z_0}$ corresponding to the base feature vector, $F_{Z_{50}}$ corresponding to prospective feature vector that 'look ahead' 50 samples (=1 second), and $F_{Z_{-50}}$ a retrospective feature vector that 'look behind' 50 samples. These feature modifications for the lag models ($Z_{50}$ and $Z{-}50$) were performed by trial to avoid across trial confounds. Because the best performing lag model was the $Z_{50}$ 'look ahead' encoder, all additional encoder models were trained with this zone feature scheme and therefor referred to the $Z$ model. Cue encoding is modeled in two ways. First as a global signal that does not interact with the feature vector (a rate-remapping model, $F_{Z+C}$). This was implemented with two additional inputs into the model, one for each cue, with value of 1 for the samples that corresponded to each trial type and zero otherwise (n of features = 41). Of note, because only one of the cues was ever present on a given trial, it was possible to

model the cue with a single binary feature. This produced the qualitative results and predictions with one fewer degree of freedom, but zone coefficients then required an extra transformation for interpretation. Having each cue modeled separately produced immediately interpretable cue and zone coefficients. In the second modeling approach, each position of the maze was modeled as a function of cue (interactive model or global-remapping model, $F_{ZxC}$), that is - a second set of 39 features were added to model, with each 39 position set only being active on trials that correspond to each cue type (*Figure 3—figure supplement 1*). After fitting Least Squares Regression for the base model ($Z_0$, 0 lag, no cue signal), we obtained the weights $\hat{E}_{Z_0}$ such that the predicted firing rate could be represented as $\hat{V}_{Z_0} = \hat{E}_{Z_0} \cdot F_{Z_0}$. Where firing rate used to fit the data can then be expressed as $V = \hat{V}_{Z_0} + \Xi_{Z_0}$, where $\Xi_{Z_0} \sim \mathcal{N}_{N_u}(0, \Sigma)$. The $N_u$ represents the number of fitted units. The coefficient of determination ($R^2$) was computed for each neuron and test fold, with the average across folds used to summarize the extent to which a given encoding model predicted firing rate ($\hat{v}_{u_i}$) the firing rate of each neuron ($v_{u_i}$), where $u_i$ represents unit $i$.

$$R^2_{u_i} = 1 - \frac{\mathbf{1}^T \cdot (vn_i - \hat{v}_{u_i})^2}{\mathbf{1}^T \cdot (v_{u_i} - \bar{v}_{u_i})^2}$$

where $\bar{v}_{u_i}$ represents the mean firing rate for unit $i$, and 1 is a vector of ones. On test data, $R^2$ values can be negative as a model can be arbitrarily worse than the mean in estimating the firing rate fluctuations. However, values less than negative one reflected convergence issues and were excluded from further analyses (# unit exclusions by analysis: cue 32 units, RW 27 units).

## Decoder

A decoder fitted with Multinomial Logistic Regression can be trained to predict position in the maze by time sample, where the probability of being in zone $z[t]$ given population neural activity $V[t]$ can be expressed as:

$$P\left(z[t]|V[t]\right) = \Phi(\hat{D}_1 \cdot V[t], ..., \hat{D}_z \cdot V[t], ..., \hat{D}_{N_Z} \cdot V[t]) \tag{7}$$

where $\hat{D}_z$ represents the model's matrix of weights that map population activity to maze position $z$, and $\Phi$ is the softmax function (multi-class version of the logistic function). This decoder provides probability values for all positions at a given time sample. Other variables of interest have single output by trial, namely cue, decision and goal. In this work, we employed a decision decoder, which can be represented as:

$$P(Decision = Left|V[t]) = \frac{1}{1 + e^{-(\hat{D}_{Left} \cdot V[t])}}$$

Note that the decoder outputs a sample wise estimates of each of the trial-wise variables of interest, in this case the subject's decision. To quantify performance, we take the average probability output by zone across samples by trial, producing a matrix of trials x zones. To get a more continuous estimate, probabilities are converted to logits and integrated across zones (*Figure 3*, *Figure 3—figure supplement 1*).

## Open-field tuning metrics

Traditional open-field tuning metrics were quantified as in previous studies (e.g. *Hardcastle et al., 2017b*). Briefly, speed-score corresponded to the linear correlation between binned speeds and firing rate activity by unit. Head-direction scores were computed as $R = \frac{1}{N}\Sigma_i^N fr_i e^{j*\theta_i}$, where $N$ is the number of samples, $i$ is the sample number, $\theta$ is the head direction of the animal, and $j = \sqrt{-1}$. Periods of immobility were excluded from this computation (speed <2 cm/s), and to obtain a final score the magnitude of $R$ is taken. Finally, position scores corresponded to the correlation between spatial rate maps corresponding to the first half and second half of the experimental sessions.

## Open-field encoding models

Three main models were used to quantify the extent of spatial/navigational coding of single units: position, head-direction, and speed. Blocked time-series cross validation was followed to generate training and test sets for the models, with each block being of length of 20 s. Periods of immobility

speed $< 2cm/second$ were excluded from the head direction models. Binned head-direction and speed behavioral features were used to model those variables. The position model also employ binned positions, but then transformed into a lower dimensional space using PCA. All models predicted firing-rate using simple linear regression. Quantification on test sets was performed by comparing the test firing rate and corresponding predicted rates and using $R^2$ and spatial map correlations $r_p$. Aggregate models took training set predicted firing rates for each model (speed, head-direction and position) as features (3 total features), and was correspondingly tested on unseen test data. The coefficients of the aggregate models were used to cluster these cells into functional coding groups.

## Acknowledgements

This work was supported by an NIMH F32 Fellowship (1F32MH119766) to AG, the Office of Naval Research (N00141812690), the Simons Foundation (SCGB 542987SPI), NIMH (MH126904), the Vallee Foundation, and the James S McDonnell Foundation to LMG. We would like thank Loren M Frank for initial discussions on the adaptation and building of the maze, David Jessen and Ashley Henderson for subject handling and training, and Adriana Diaz for animal husbandry and lab support.

## Additional information

### Competing interests

Lisa M Giocomo: Reviewing editor, *eLife*. The other author declares that no competing interests exist.

### Funding

| Funder | Grant reference number | Author |
| --- | --- | --- |
| National Institute of Mental Health | 1F32MH119766 | Alexander Gonzalez |
| National Institute of Mental Health | MH126904 | Lisa M Giocomo |
| Office of Naval Research | N00141812690 | Lisa M Giocomo |
| Simons Foundation | SCGB 542987SPI | Lisa M Giocomo |
| Vallee Foundation | | Lisa M Giocomo |
| James S. McDonnell Foundation | | Lisa M Giocomo |

The funders had no role in study design, data collection and interpretation, or the decision to submit the work for publication.

### Author contributions

Alexander Gonzalez, Conceptualization, Resources, Data curation, Software, Formal analysis, Funding acquisition, Validation, Investigation, Visualization, Methodology, Writing - original draft, Project administration, Writing - review and editing; Lisa M Giocomo, Conceptualization, Resources, Supervision, Funding acquisition, Writing - original draft, Project administration, Writing - review and editing

### Author ORCIDs

Alexander Gonzalez ⓘ http://orcid.org/0000-0002-7328-9533
Lisa M Giocomo ⓘ http://orcid.org/0000-0003-0416-2528

### Ethics

All techniques were approved by the Institutional Animal Care and Use Committee at Stanford University School of Medicine (IACUC) under protocol #27694.

Reviewer #1 (Public Review): https://doi.org/10.7554/eLife.85646.3.sa1
Reviewer #2 (Public Review): https://doi.org/10.7554/eLife.85646.3.sa2
Reviewer #3 (Public Review): https://doi.org/10.7554/eLife.85646.3.sa3

Author Response https://doi.org/10.7554/eLife.85646.3.sa4

## Additional files

### Supplementary files
• MDAR checklist

### Data availability
Code is available for download as a repository on Github https://github.com/alexgonzl/TMA, (copy archived at *Gonzalez, 2023*). All data required to reproduce the paper figures is available through links in the Github repository.

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
