## [Editor Report · eLife assessment]

In this study, neurons were recorded and combined across the parahippocampal area while rats performed a memory-guided spatial navigation task. Sophisticated analytical tools were used to provide **convincing** evidence that neuronal populations in these areas show behavior-related changes that might indicate the encoding of errors by the system. The **valuable** results suggest that rate remapping is a likely mechanism to support changes in representations that support memory-guided behavior in these regions, most interestingly in neurons that code head direction.

---

## [Referee Report · Reviewer #1 (Public Review)]

In this study, single neurons were recorded, using tetrodes, from the parahippocampal cortex of 5 rats navigating a double-Y maze (in which each arm of a Y-maze forks again). The goal was located at any one of the 4 branch terminations, and rats were given partial information in the form of a light cue that indicated whether the reward was on the right or left side of the maze. The second decision point was un-cued and the rat had no way of knowing which of the two branches was correct, so this phase of the task was more akin to foraging. Following the outbound journey, with or without reward, the rat had to return (inbound journey) to the maze start, to begin again.

Neuronal activity was assessed for correlations with multiple navigation-relevant variables including location, head direction, speed, reward side, and goal location. The main finding is that a high proportion of neurons showed an increase in firing rate when the animal made a wrong turn at the first branch point (the one in which the correct decision was signalled). This increase, which the authors call rate remapping, persisted throughout the inbound journey as well. It was also found that head direction neurons (assessed by recording in an open field arena) in the same location in the room were more likely to show the rate change. The overall conclusion is that "during goal-directed navigation, parahippocampal neurons encode error information reflective of an animal's behavioral performance" or are "nodes in the transmission of behaviorally relevant variables during goal-directed navigation."

Overall I think this is a well-conducted study investigating an important class of neural representation: namely, the substrate for spatial orientation and navigation. The analyses are very sophisticated - possibly a little too much so, as the basic findings are relatively straightforward and the analyses take quite a bit of work to understand. A difficulty with the study is that it was exploratory (observational) rather than hypothesis-driven. Thus, the findings reveal correlations in the data but do not allow us to infer causal relationships. That said, the observation of increased firing in a subset of neurons following an erroneous choice is potentially interesting. However, the effect seems small. What were the actual firing rate values in Hz, and what was the effect size?

I also feel we are lacking information about the underlying behavior that accompanies these firing rate effects. The authors say "one possibility is that the head-direction signal in the parahippocampal region reflects a behavioral state related to navigational choice or the lack of commitment to a particular navigational route" which is a good thought and raises the possibility that on error trials, rats are more uncertain and turn their heads more (vicarious trial and error) and thus sample the preferred firing direction more thoroughly. Another possibility is that they run more slowly, which is associated with a higher firing rate in these cells. I think we therefore need a better understanding of how behaviour differed between error trials in terms of running speed, directional sampling, etc. A few good, convincing raw-data plots showing a remapping neuron on an error trial and a correct trial on the same arm would also be helpful (the spike plots were too tiny to get a good sense of this: fewer, larger ones would be more helpful). It would be useful to know at what point the elevated response returned to baseline, how - was it when the next trial began, and was the drop gradual (suggesting perhaps a more neurohumoral response) or sudden?

Comments on the revised submission:

The authors have clarified a number of points arising from my original review but some remain.

On the issue of hypotheses: I was really referring, and apologies that I was unclear on this, to the hypothesis about the neural responses predicted in this experiment. The authors aimed to "examine whether spatial representations flexibly adapt to behaviorally relevant factors" but this is not really a hypothesis as such, in the true mechanistic sense so much as "let's see what we can find" which is not an invalid reason to do this type of study. However, no manipulations were made that test causal relationships arising from the study. It thus remains observational. It does however raise testable hypotheses which is valuable. The strongest in my mind is that the rise in firing rates is a catecholamine response to frustration, a conclusion supported by the slow temporal dynamics of the changes.

On the issue of running speed: it needs to be ruled out that this might have been the cause of the altered firing rates since running speeds were different. More generally, the lack of other concurrent behavioral data means we cannot rule out other possible behavioral bases to this effect that are unrelated to error but are related to the motor correlates of the error.

---

## [Referee Report · Reviewer #2 (Public Review)]

This work recorded neurons in the parahippocampal regions of the medial entorhinal cortex (MEC) and pre- and para-subiculum (PrS, PaS) during a visually guided navigation task on a 'tree maze'. They found that many of the neurons reflected in their firing the visual cue (or the associated correct behavioral choice of the animal) and also the absence of reward in inbound passes (with increased firing rate). Rate remapping explained best these firing rate changes in both conditions for those cells that exhibited place-related firing. This work used a novel task, and the increased firing rate at error trials in these regions is also novel.

The limitation is that cells in these regions were analyzed together.

Comments on the revised submission:

I accept the authors' response that histological differentiation of these regions was not possible.

---

## [Referee Report · Reviewer #3 (Public Review)]

Summary & Strengths:

This study is useful in revealing the neural correlates of goal-directed navigation in the rodent parahippocampal regions, including the medial entorhinal cortex, presubiculum, and parasubiculum. It shows that task-relevant information represented by the parahippocampus is strongly related to task performance. It also reports the relationships of navigational factors (e.g., head direction signal) recorded during foraging in an open field with task variables.

Gonzalez and Giocomo investigated the neural activities in the parahippocampal cortex modulated by visual cues and error signals while the animal performed a goal-directed navigation task on the tree maze. They confirmed that the firing rates and spatial firing patterns in the parahippocampus were significantly correlated with the animal's task performance and the general navigational coding in the open field arena. The authors have concluded that the parahippocampal neurons encode mismatch-like signals, suggesting the functional role of the parahippocampus as a feedback system in a goal-directed task. However, a few major concerns should be addressed more closely to support the conclusion.

1. Due to the limitations of histological verification, the neural responses in the medial entorhinal cortex, presubiculum, and parasubiculum are analyzed together, and this limits the study from understanding the differential information processing across these regions. Because the medial entorhinal cortex and the pre/parasubiculum are believed to be located in very different positions in the information flow within the rodent medial temporal lobe with different anatomical connections, it would have been more convincing if the distinctive functions between the regions could be identified.

2. The authors should carefully differentiate rate remapping and global remapping in their analysis. Rate remapping generally indicates firing rate modulation with little or no shift of spatial firing fields (Leutgeb et al., 2005; Colgin et al., 2008). Therefore, the neurons exhibiting global remapping should not be included in the analysis suited for rate remapping (e.g., the encoding model that considers the cue-dependent rate-remapping effect).

3. One of the major findings in this study is that the parahippocampal neural responses to a visual cue or reward were correlated with task performance. One can expect that cue representation before the decision point is likely to have a greater impact on task performance. Although the Uz score between the left cue and right cue seemed not significantly different from zero on the stem, it would be beneficial if the authors verify whether the remapping score based on the firing rate maps will still be correlated with the task performance when examined only before the decision point, not for the entire maze.

4. There is a need to set the analytic epoch in more detail. The boundary between outbound and inbound journeys was set as 'last goal well visit.' However, even in a correct trial, if the reward was not received in the first goal well, an error signal could occur before the animal triggered the second goal well which was rewarding. This might have caused the rate remapping between two cue conditions, specifically on the arms. To eliminate this possibility, it is recommended to set the outbound journey from the home well trigger to the first goal well approach or to select only trials where the animal received rewards from the first goal well triggering.

Weaknesses:

Incomplete results could limit support for the arguments of the study and may require more rigorous analytical methods.

---

## [Author Response]

The following is the authors’ response to the original reviews.

**Reviewer #1 (Public Review):**
In this study, single neurons were recorded, using tetrodes, from the parahippocampal cortex of 5 rats navigating a double-Y maze (in which each arm of a Y-maze forks again). The goal was located at any one of the 4 branch terminations, and rats were given partial information in the form of a light cue that indicated whether the reward was on the right or left side of the maze. The second decision point was uncued and the rat had no way of knowing which of the two branches was correct, so this phase of the task was more akin to foraging. Following the outbound journey, with or without reward, the rat had to return (inbound journey) to the maze and start to begin again.Neuronal activity was assessed for correlations with multiple navigation-relevant variables including location, head direction, speed, reward side, and goal location. The main finding is that a high proportion of neurons showed an increase in firing rate when the animal made a wrong turn at the first branch point (the one in which the correct decision was signalled). This increase, which the authors call rate remapping, persisted throughout the inbound journey as well. It was also found that head direction neurons (assessed by recording in an open field arena) in the same location in the room were more likely to show the rate change. The overall conclusion is that "during goal-directed navigation, parahippocampal neurons encode error information reflective of an animal's behavioral performance" or are "nodes in the transmission of behaviorally relevant variables during goal-directed navigation."Overall I think this is a well-conducted study investigating an important class of neural representation: namely, the substrate for spatial orientation and navigation. The analyses are very sophisticated - possibly a little too much so, as the basic findings are relatively straightforward and the analyses take quite a bit of work to understand. A difficulty with the study is that it was exploratory (observational) rather than hypothesis-driven. Thus, the findings reveal correlations in the data but do not allow us to infer causal relationships.

We would like to clarify that this report consists of hypothesis-driven experiments, with post-hoc exploratory analyses. We have now made hypotheses more explicit in the text, and pointed out that follow-up analyses were to understand how these effects came to be. We thank the reviewer for pointing out that our hypotheses were not explicit in the introduction. We believe our results open the door for investigating the causal role of these regions in the propagation or generation of error signals during navigational behavior. Those types of experiments are however, outside the scope of the current work.

That said, the observation of increased firing in a subset of neurons following an erroneous choice is potentially interesting. However, the effect seems small. What were the actual firing rate values in Hz, and what was the effect size?

We thank the reviewer for the opportunity to clarify the effect size question. As there are multiple neurons in the analyses, differences in firing rate need necessarily to be normalized by a neuron's mean activity. For example, a difference of 1 spk/s is less meaningful when a neuron's base rate is 50 spk/s than when it is 10spks/s. Furthermore, our reports are for population level analyses, at which point comparing raw firing rate values and differences becomes more challenging. Nonetheless, we are including these raw metrics in two new supplemental figures (Figure 2 - figure supplement 4,5), where differences in individual neurons change can be up to 15 spks/s. Additionally, the patterns and statistical results observed in the main text are preserved, with outbound Right Cue minus Left Cue showing a left>stem>right (indicating error coding), and RW minus NRW showing negative values across all segments, indicating NRW>RW or higher activity following on inbound unrewarded trials. Statistics follow the corresponding main text results (Cue: segment LRT = 71.70; RW: segment LRT=45.80).

I also feel we are lacking information about the underlying behavior that accompanies these firing rate effects. The authors say "one possibility is that the head-direction signal in the parahippocampal region reflects a behavioral state related to the navigational choice or the lack of commitment to a particular navigational route" which is a good thought and raises the possibility that on error trials, rats are more uncertain and turn their heads more (vicarious trial and error) and thus sample the preferred firing direction more thoroughly. Another possibility is that they run more slowly, which is associated with a higher firing rate in these cells. I think we, therefore, need a better understanding of how behavior differed between error trials in terms of running speed, directional sampling, etc.

In terms of running speed, there was a small effect of mean running speed between correct and incorrect trials (across subjects LMEM: Cue correct>incorrect Z=2.3, p=0.02; RW Z=2.15, p=0.03). In most neurons, increases in speed will be accompanied by increases in firing rate. Thus, the differences in running speed cannot explain the observed results, as higher speed during correct trials would predict higher activity, which is the opposite of what we found.

A few good, convincing raw-data plots showing a remapping neuron on an error trial and a correct trial on the same arm would also be helpful (the spike plots were too tiny to get a good sense of this: fewer, larger ones would be more helpful).

Additional plots for individual units have been added, Figure 2 - figure supplement 3.

It would be useful to know at what point the elevated response returned to baseline, how - was it when the next trial began, and was the drop gradual (suggesting perhaps a more neurohumoral response) or sudden.

Due to the experimental design, this question cannot be addressed fully. Concretely, error trials incur a time-penalty in which the rats need to wait an additional 10 seconds before the next trial, while a new trial would start immediately when the animal nose-poked the home well after a correct trial. Nonetheless, the data on Reward provides insight into this question. The magnitude of the responses on left and right segments of the maze were larger than on the stem for Unrewarded (NRW) vs Rewarded (RW) trials on inbound trajectories, Fig. 4c. This suggests that while activity is still elevated post-incorrect throughout the maze, across units, this effect is smaller on the stem segment. Additionally, the analyses indicate that in the transition between outbound vs inbound trajectories (Figure 4 - figure supplement 3), activity patterns are sustained (incorrect>correct). Together, these results indicate that elevated "error-like" signal are slow in returning to baseline.

**Reviewer #2 (Public Review):**
This work recorded neurons in the parahippocampal regions of the medial entorhinal cortex (MEC) and pre- and para-subiculum (PrS, PaS) during a visually guided navigation task on a 'tree maze'. They found that many of the neurons reflected in their firing the visual cue (or the associated correct behavioral choice of the animal) and also the absence of reward in inbound passes (with increased firing rate). Rate remapping explained best these firing rate changes in both conditions for those cells that exhibited place-related firing. This work used a novel task, and the increased firing rate at error trials in these regions is also novel. The limitation is that cells in these regions were analyzed together.

We acknowledge this limitation on our study, and we believe there might be interesting differences between these regions. Unfortunately, the post-mortem extraction of the recording implant micro-drive used for these experiments generated too much tissue damage for exact localization of the tetrodes. Nonetheless, given that the patterns were observed in all subjects, we are confident that at least the major findings of "error-like" signaling is present across the parahippocampal regions. At the same time, the distributions of functional cell types as defined in the open field are different across the PaS, PrS and MEC, leaving the possibility of a more nuanced error coding scheme by region.

**Reviewer #3 (Public Review):**
The authors set out to explore how neurons in the rodent parahippocampal area code for environmental and behavioral variables in a complex goal-directed task. The task required animals to learn the association between a cue and a spatial response and to use this information to guide behavior flexibly on a trial-by-trial basis. The authors then used a series of sophisticated analytical techniques to examine how neurons in this area encode spatial location, task-relevant cues, and correct vs. incorrect responding. While these questions have been addressed in studies of hippocampal place cells, these questions have not been addressed in these upstream parahippocampal areas.Strengths:1. The study presents data from ensembles of simultaneously recorded neurons in the parahippocampal region. The authors use a sophisticated method for ensuring they are not recording from the same neurons in multiple sessions and yet still report impressive sample sizes.1. The use of the complex behavioral task guards against stereotyped behavior as rats need to continually pay attention to the relevant cue to guide behavior. The task is also quite difficult ensuring rats do not reach a ceiling level of performance which allows the authors to examine correct and incorrect trials and how spatial representations differ between them.1. The authors take the unusual approach of not pre-processing the data to group neurons into categories based on the type of spatial information that they represent. This guards against preconceived assumptions as to how certain populations of neurons encode information.1. The sophisticated analytical tools used throughout the manuscript allow the authors to examine spatial representations relative to a series of models of information processing.1. The most interesting finding is that neurons in this region respond to situations where rewards are not received by increasing their firing rates. This error or mismatch signal is most commonly associated with regions of the basal ganglia and so this finding will be of particular interest to the field.Weaknesses:1. The histological verification of electrode position is poor and while this is acknowledged by the authors it does limit the ability to interpret these data. Recent advances have enabled researchers to look at very specific classes of neurons within traditionally defined anatomical regions and examine their interactions with well-defined targets in other parts of the brain. The lack of specificity here means that the authors have had to group MEC, PaS, and PrS into a functional group; the parahippocampus. Their primary aim is then to examine these neurons as a functional group. Given that we know that neurons in these areas differ in significant ways, there is not a strong argument for doing this.

See response to Reviewer 2.

1. The analytical/statistical tools used are very impressive but beyond the understanding of many readers. This limits the reader's ability to understand these data in reference to the rest of the literature. There are lots of places where this applies but I will describe one specific example. As noted above the authors use a complex method to examine whether neurons are recorded on multiple consecutive occasions. This is commendable as many studies in the field do not address this issue at all and it can have a major impact as analyses of multiple samples of the same neurons are often treated as if they were independent. However, there is no illustration of the outputs of this method. It would be good to see some examples of recordings that this method classifies as clearly different across days and those which are not. Some reference to previously used methods would also help the reader understand how this new method relates to those used previously.

We have added an additional Supplemental Figure 7 - figure supplement 1, that showcases the matching waveform approach. In the original manuscript, Fig. 7a provided an example output of the method.

1. The effects reported are often subtle, especially at the level of the single neuron. Examples in the figures do not support the interpretations from the population-level analysis very convincingly.

Additional plots for individual units have been added, Figure 2 - figure supplement 3. However, the effects, though small by unit, are consistent across neurons and subjects.

The authors largely achieve their aims with an interesting behavioral task that rats perform well but not too well. This allows them to examine memory on a trial-by-trial basis and have sufficient numbers of error trials to examine how spatial representations support memory-guided behavior. They report ensemble recordings from the parahippocampus which allows them to make conclusions about information processing within this region. This aim is relatively weak though given that this collection of areas would not usually be grouped together and treated as a single unitary area. They largely achieve their aim of examining the mechanisms underlying how these neurons code task-relevant factors such as spatial location, cue, and presence of reward. The mismatch or error-induced rate remapping will be a particularly interesting target for future research. It is also likely that the analytical tools used in this study could be used in future studies.
**Reviewer #1 (Recommendations For The Authors):**
1. Typo: "attempted to addresses these challenges"

We thank the reviewer for pointing out, this has been fixed.

1. "classified using tuning curve based metrics" - what does "tuning curve" mean in this context?

We have clarified this sentence in the main text.

1. "MEC neurons encode time-elapsed" should be "MEC neurons encode time elapsed" (no hyphen)

We thank the reviewer for pointing out, this has been fixed.

1. "a phenomenon referred to as 'global remapping'." - I dislike this term because it has two meanings in the literature. On the one hand, it is used to contrast with rate remapping: that is, it refers to a change in the location of place fields. On the other hand, it refers to the remapping of the whole population of cells at once, as contrasted with partial remapping. I suggest calling them location remapping (vs. rate) and complete remapping (vs. partial)

We agree that this is an overloaded term in the field. We have added 'location remapping' in the intro as a more specific term for global remapping.

1. " tasks with no trial-to-trial predictability or experimenter-controlled cues and goals in the same environment." - ambiguously worded as it isn't clear whether the "no" refers to one or both of what follows. Also needs a hyphen after experimenter.

We thank the reviewer for pointing out, this sentence has been reworded for clarity.

1. " neurons changed their firing activity as a function of cue identity" - this is confounded by behavior and reward contingency, both linked to cue identity, so the statement is slightly misleading.

We thank the reviewer for pointing this out, however, we disagree that this wording is misleading. Neurons changed their activity as a function cue identity and reward contingencies. Why neurons change their activity in such a manner is a different, more nuanced question, that we addressed through our analyses by converging on the "error" like signal that these signals seem to carry.

1. "remapping" - I am not fully comfortable with the use of this term in this context. It derives from the original reports of change in the firing location of place cells, and the proposal that these cells form a "map" with the change in activity reflecting recruitment of a new map. With observations of rate changes in some place cells, the new term "rate remapping" was introduced, and now the authors are using "rate remapping" to mean firing rate changes in non-spatial cells. The meaning is thus losing its value. "Re-coding" might be slightly better, although we can argue about whether "code" is much better than "map"

While we agree with the reviewer that "remapping" has been coerced into a grab-all term, these are the accepted semantics in the field. Thus, we are disinclined to change the language.

1. Figure 1 - it would be useful to indicate somehow that one of the decision points was cued and once free choice with the random outcome - it took me a while to work this out. Also, the choice of colors for the cues needs explaining - my understanding is that rats are very insensitive to these wavelengths. And what does Pse mean? I didn't understand those scatterplots at all.

The section "Tree-Maze behavior and electrophysiological recordings" under Results go into the details of the task. A reference and additional context for the selection of cues is now included in the "Behavioral Training" methods section. Rats possess dichromatic vision systems. Caption of Figure 1, 2 includes what Pse means, the performance of a subject for a given session. The scatter plots relate remapping to performance.

1. Also on Figure 1 - in the examples shown, it looks like the animals always checked the two end arms in the same order. Was this position habit typical?

We have added additional context in "Behavioral Training" methods section. Well trained rats do exhibit stereotyped behaviors (eg. going to one well then the other).

1. "...we hypothesized that the cue remapping score would be related to a subject's performance in the task." I am struggling to see why this doesn't follow trivially from the observation that remapping occurred on error trials.

We thank the reviewer for pointing out that this could use further clarity. We have added that the magnitude of remapping is what should relate to performance. To further clarify, remapping does not occur on error trials, remapping as operationally defined in this work, is the difference of spatial maps as a function of Cue identity or Reward contingency. Thus, as a difference metric, remapping occurs because there is a difference in activity between correct and incorrect trials. The magnitude of that difference need not relate to how the subject performed on the task.

1. "With this approach, found that incorrect coding units were more likely to overlap between cue and RW coding units than correct." Missing "we". I didn't understand this sentence - what does "overlap" mean?

We have added a sentence to further clarify this point.

1. "We found that incorrect>correct activity levels on outbound trajectories predicted incorrect>correct activity levels on inbound trajectories" - I don't understand how this can be the case given that many of these units were head direction tuned and therefore shouldn't even have been active in both directions.

As seen in Figure 7b, we were able to match 217 units across tasks. Of those, "Cluster 0" with 98 units showed strong head-direction coding. While "Cluster 0" units showed strong remapping effects, there were a lot of other units that could have contributed to the "incorrect>correct" across (out/in)-bound segments. Further, head-direction coding is defined in the Open-field environment, and there's no constraint on what these neurons could do on the Tree Maze task.

13). " Error or mismatch signals conform a fundamental computation" - should be "perform"

Wording slightly changed, but "conform" as in "act in accordance to" is what we intend here.

1. " provides it with the required stiffness and chemical resistivity"- what does "chemical resistivity" mean? To what chemicals?

This is mostly in reference to rat waste and cleaning products (alcohol). We changed the wording to durability for simplicity.

1. Supp Fig 1 shows that behavioral performance was very distinctly different for one of the animals. Was its neural data any different? What happens to the overall effect if this animal is removed from the analysis?

Unless otherwise stated, all analyses are performed through linear mixed effects with "subject" as a random effect. Thus, the effects of individual subjects are accounted for.

1. Histology - it would be useful to have a line drawing from the atlas alongside the micrographs to enable easier anatomical understanding.

There was variability in the medial lateral location of the tetrodes across animals and in the histological images provided and thus, we felt this would not be useful information as a single line drawing will not encompass/apply to all histology photos.

1. Supp. Fig. 5/6 I didn't understand what Left, Stem, and Right mean at the top. Also, the color keys are too tiny to be noticed

An additional sentence has been added to the caption to clarify that left, stem, right refer to what segment was selected via the ranking of scores.

**Reviewer #2 (Recommendations For The Authors):**
Was there a particular reason why cells in these regions were analyzed together? Can some of the results be tested for cells of a particular region, especially the MEC? One major limitation of these results is that it is unclear which regions it applies to, e.g., one cannot be certain that data shows here that MEC cells have these firing properties.

Damage due to the extraction of the recording tetrode bundle was extensive and we were not able to parcelate out individual regions. We have added additional details on this in the "Histology" section of the methods.

It is unclear how many cells in each region are included in each analysis. There is supplementary fig 3 but not sure how many of these met the criteria to be included in a certain analysis and it does not differentiate regions. Also, was any of the MUA included in the analyses?

Isolated single units were included in all analyses, but we did not differentiate from what region each unit came from. Analyses that include MUA are separate from the main findings, and are included in supplemental figures as reference.

Was the error trial defined as a trial when the animal did not make the right light-guided choice or did it also include cases in which the light-related arm choice was correct, but the animal first went to the unrewarded end arm? Nomenclature in the results is not explained well - what is an unrewarded trial or unrewarded trajectory or an error trial?

We have added a new paragraph in the methods under Behavioral Training that address trial nomenclature. This methods section is now referenced twice in the initial paragraphs of the results section.

Were any grid cells included in the data, especially could any cross-matched across the open field and the maze runs?

This was indeed a question of interest to us, however, the number of grid-cells recorded was not adequate for meaningful statistical inference. We further sought to avoid tuning curve based functional classifications of units.

In general, the results section is difficult to read, and its accessibility could be improved.

We thank the reviewer for this accessibility point. We hope that the small tweaks as a product of this revision will improve the readability of the manuscript. We tried to have major takeaways for each result, but the nature of the analyses necessarily make the text somewhat dense.

Minor:One of the Figure 3f references should be Figure 3g and later, Figure 44 should be corrected.

We thank the reviewer for noting this, it has been fixed.

**Reviewer #3 (Recommendations For The Authors):**
There are a number of issues that I think could be addressed to improve the manuscript:1. The figure could make it clearer where the LED panel is. Are the authors confident the rats see the cue on each trial?

We have added a new supplemental figure to address this question (Figure 1 - figure supplement 1). The new figures show the 3D geometry of the maze and the location of the Cue panel. The rats were able to see the cue, otherwise task performance would have remained at chance levels.

1. The same maze has been used in a series of studies of hippocampal place cells by Paul Dudchenko's group. They also went on to examine how these representations are affected in a very similar cued spatial response task. These studies should be acknowledged.

We thank the reviewer for pointing out this oversight. We have added the Ainge et al. citation ( https://doi.org/10.1523/JNEUROSCI.2011-07.2007) when first introducing the maze and in the methods.

1. In a number of supplementary figures, the authors present neurons that are selective for different properties such as segment, cue, reward, and direction. However, the number of spatially and cue-selective cells and the criteria by which cells are designated as selective are not reported. The analyses of spatial remapping and response to cues are done at the population level so I'm not sure how these cells are classified or selected for the figures.

The procedure for selection is included in the figure captions. Each unit is ranked based on the Uz score by segment as originally shown in Figures 2 and 4.

1. Related to this, the example cells on the figures do not clearly represent the effects presented. For example, given the title of Figure 2, I assume that the cells in 2B significantly remap. However, they don't look like they remap - the cells in the top row show rate remapping in one segment of the maze while the cells in the bottom do not show clear rate remapping responses. I suspect that traditional rate map-based analyses using maps based on consistently sized pixels rather than large segments would show only very modest changes in correlations or rates across these different types of trials. It is important to report the findings in this way as the authors interpret their data relative to the rate-remapping studies which have used these analyses. Readers who do not have the time or expertise to examine the methods in detail will conclude that the effects reported here are the same as previous rate remapping studies which the examples suggest is not the case.

Additional plots for individual units have been added to the supplement, Figure 2 - figure supplement 3. However, the effects, though small by unit, are consistent across neurons and subjects (Figure 2 - figure supplement 5).

1. Why is there a bias on the stem in 2C? This is of similar size to the effect on the right size and so deserves discussion.

The analysis in question is the across unit level bias in cue-coding by maze segment. The left segment shows elevated Right Cue coding, while the right segment shows elevated Left Cue coding. There was one reported statistical result, the main effect of segment in the Linear Mixed Effects model. We expand this result in the following two ways:

1. Individual statistical results by segment

a. Left Segment (Uz Coef. Estimate = 0.5, CI95%=[0.26, 0.75; p<1e-4])

b. Stem Segment (Uz Coef. Estimate = 0.22, CI95%=[-0.01, 0.47]; p=0.06)

c. Right Segment (Uz Coef. Estimate = -0.27, CI95%=[-0.51, -0.03], p=0.03)

1. Reporting the joint hypothesis test of left > stem > right by unit.

a. X2=90.45, p=2.28e-20

b. The comparison of left>stem by unit:

i. coefficient estimate = 0.28, CI95%=[0.11, 0.44], p=0.0008

Although the reviewer is correct in pointing out the effect size similarity, the appropriate statistical comparisons within and across units support the stated conclusions. In terms of systematic coding bias, there is a small bias across units (60% of units) and animals (4 out 5) for the Right Cue. Although interesting, this effect is orthogonal to the comparisons of interests (within unit differences). In order to highlight this point we have added the statistics of the joint hypothesis test of left>stem>right to the main manuscript.